# Disentangling the effects of structure and lone-pair electrons in the lattice dynamics of halide perovskites

Sebastián Caicedo-Dávila [1], Adi Cohen[2], Silvia G. Motti[3,4], Masahiko Isobe [5], Kyle M. McCall [6,7,10], Manuel Grumet[1], Maksym V. Kovalenko [6,7], Omer Yaffe[2], Laura M. Herz [3,8], Douglas H. Fabini [5,9] ✉ & David A. Egger [1] ✉

Halide perovskites show great optoelectronic performance, but their favorable properties are paired with unusually strong anharmonicity. It was proposed that this combination derives from the $ns^2$ electron configuration of octahedral cations and associated pseudo-Jahn–Teller effect. We show that such cations are not a prerequisite for the strong anharmonicity and low-energy lattice dynamics encountered in these materials. We combine X-ray diffraction, infrared and Raman spectroscopies, and molecular dynamics to contrast the lattice dynamics of $CsSrBr_3$ with those of $CsPbBr_3$, two compounds that are structurally similar but with the former lacking $ns^2$ cations with the propensity to form electron lone pairs. We exploit low-frequency diffusive Raman scattering, nominally symmetry-forbidden in the cubic phase, as a fingerprint of anharmonicity and reveal that low-frequency tilting occurs irrespective of octahedral cation electron configuration. This highlights the role of structure in perovskite lattice dynamics, providing design rules for the emerging class of soft perovskite semiconductors.

Halide perovskites (HaPs) with formula $AMX_3$ generated enormous research interest because of their outstanding performance in optoelectronic devices, most notably in efficient solar cells[1–3]. These compounds are highly unusual among the established semiconductors because they feature an intriguing combination of properties. Strong anharmonic fluctuations[4–6] in these soft materials appear together with optoelectronic characteristics that are favorable for technological applications[7,8]. This confluence raised puzzling questions regarding the microscopic characteristics of the materials and the compositional tuning of their properties alike. On the one hand, the soft anharmonic

nature of the HaP structure may be beneficial in self-healing mechanisms of the material[9–11], allowing for low-energy synthesis routes in their fabrication. On the other hand, pairing of anharmonic fluctuations and optoelectronic processes for key quantities of HaPs, *e.g.*, band gaps[12–15], optical absorption profiles[16–18], and charge-carrier mobilities[8,19–25], exposed incomplete microscopic rationales for the fundamental physical processes involved in solar-energy conversion. Established materials design rules are now being challenged by these observations, opening a gap in our protocols for making improved compounds.

[1]Physics Department, TUM School of Natural Sciences, Technical University of Munich, Garching, Germany. [2]Department of Chemical and Biological Physics, Weizmann Institute of Science, Rehovot, Israel. [3]Clarendon Laboratory, Department of Physics, University of Oxford, Oxford, UK. [4]School of Physics and Astronomy, Faculty of Engineering and Physical Sciences, University of Southampton, Southampton, UK. [5]Max Planck Institute for Solid State Research, Stuttgart, Germany. [6]Laboratory of Inorganic Chemistry, Department of Chemistry and Applied Biosciences, ETH Zurich, Zürich, Switzerland. [7]Laboratory for Thin Films and Photovoltaics, EMPA - Swiss National Laboratories for Materials and Technology, Dübendorf, Switzerland. [8]TUM Institute for Advanced Study, Technical University of Munich, Garching, Germany. [9]Department of Chemistry, Massachusetts Institute of Technology, Cambridge, MA, USA. [10]Present address: Department of Materials Science and Engineering, University of Texas at Dallas, Richardson, TX, USA. ✉e-mail: fabini@mit.edu; david.egger@tum.de

Significant efforts are now underway to discern the chemical effects giving rise to these remarkable properties of HaPs. Because lattice dynamical and optoelectronic properties appear both to be special and coupled in unusual ways, a common origin in chemical bonding could underlie these phenomena. In this context, an interesting chemical feature is that the octahedral cations in these compounds often bear an $ns^2$ electron configuration (*e.g.*, $Pb^{2+}$ with configuration $[Xe]6s^2$), which is not present in many other semiconductors[26]. This particular aspect of HaPs leads to a strong or weak pseudo-Jahn–Teller (PJT) effect[27–29], depending on the particulars of cation and anion composition and chemical pressure. The weak case influences local structure[30–32], lattice dynamics[33] and ionic dielectric responses[26,31,34,35], while the strong case additionally results in the formation of a stereochemically-expressed electron lone pair and impacts average crystal structures[33,36–38]. The weak PJT effect associated with $6s^2$ $Pb^{2+}$ coordinated by heavy halides plays a role in optoelectronic properties of these materials: its influence on the dielectric function can modify the Coulomb screening that is relevant for small exciton binding energies, reduced recombination rates and other key properties of HaPs[39,40].

Confluences of the propensity for lone-pair formation with structural and lattice-dynamical properties were investigated in previous work exploring the chemical space of HaPs. Gao et al.[33] found an inverse relationship between the Goldschmidt tolerance factor, $t$[41], and anharmonic octahedral tilting motions. Similarly, Huang et al. varied the A-site cation to explore interrelations of chemical, structural, and dynamical effects in HaPs[35], reporting $t$-induced modulations of octahedral tiltings and lone-pair stereoactivity. A recent study by several of the present authors found that $Cs_2AgBiBr_6$ lacks some expressions of lattice anharmonicity found in other HaP variants[42]. Because every other octahedral cation ($Ag^+$, $4d^{10}$) cannot form a lone pair in this compound, this raised the possibility that changing the electron configuration of the cations may also suppress certain aspects of the lattice dynamics in HaPs. Taken together, previous work assigned a central role of the $ns^2$ electron configuration and associated PJT effect in the anharmonic lattice dynamics of HaPs in addition to their established effect on the electronic structure and dielectric screening. However, exploring the chemical space of HaPs in this way simultaneously changes their structures. Therefore, isolating the convoluted occurrences of cation lone-pair formation propensity and purely structurally-determined changes in the lattice dynamics of HaPs remained challenging, making an assessment of the precise impact of chemical bonding on anharmonicity in these soft semiconductors largely inaccessible.

Here, we address this issue and show that an $ns^2$ cation compatible with lone-pair formation is not required for the strong anharmonicity in the low-energy lattice dynamics of soft HaP semiconductors. We disentangle structural and chemical effects in the lattice dynamics of HaPs by comparing the well-known $CsPbBr_3$ with the far less studied $CsSrBr_3$. Both exhibit almost identical geometrical and structural parameters, but $CsSrBr_3$ exhibits a negligible PJT effect on the octahedral $Sr^{2+}$ site, owing to substantially weaker vibronic coupling to degenerate excited states than in the $Pb^{2+}$ case (see Supplementary Note 1), allowing separation of the effects of the $ns^2$ electron configuration and the geometry on the lattice dynamics in a direct manner. Combining electronic structure and molecular dynamics (MD) calculations with X-ray diffraction (XRD), infrared (IR) and Raman spectroscopies, we assess a key fingerprint of vibrational anharmonicity, *i.e.*, the Raman central peak, which is a broad peak towards zero frequency in the Raman spectrum resulting from diffusive inelastic scattering[26,33–35,43–45]. While the electronic structure and dielectric properties of $CsPbBr_3$ and $CsSrBr_3$ are very different, their vibrational anharmonicities are found to be remarkably similar. In particular, the crucial dynamic octahedral tiltings giving rise to the Raman central peak are still present even in the absence of $ns^2$ octahedral cations in

$CsSrBr_3$. Our results provide microscopic understanding of precisely how the propensity for lone-pair formation influences the anharmonic octahedral tiltings that dynamically break the average cubic symmetry in both compounds, and rule out the weak PJT effect associated with the $ns^2$ main-group cations as the sole reason for the appearance of such anharmonicity in soft HaPs. These findings are important for chemical tuning of HaPs needed for new materials design.

## Results
### Electronic structure and bonding

We first investigate the electronic structure and bonding of $CsPbBr_3$ and $CsSrBr_3$ using density-functional theory (DFT). Figure 1 shows their band structure, total and projected density of states (DOS), as well as the total and projected crystal-orbital Hamilton population (COHP) of the high-temperature cubic phases of $CsPbBr_3$ and $CsSrBr_3$. The electronic band structure and bonding of $CsPbBr_3$ were extensively investigated before[46]: the conduction band minimum (CBM) is formed by anti-bonding interactions (positive COHP in Fig. 1c) between Pb-$6p$ and Br-$4p$/Br-$4s$ orbitals, while the valence band maximum (VBM) is formed by anti-bonding interactions between Br-$4p$ and Pb-$6s$ orbitals.

The electronic structure of $CsSrBr_3$ exhibits entirely different characteristics[26,47], especially a much larger band gap and weaker covalent interactions. Notably, the magnitude of the COHP is significantly reduced with respect to that of $CsPbBr_3$, indicating much greater ionicity, and the COHP is almost entirely recovered by interactions between Cs and Br. Importantly, all bands derived from anti-bonding interactions between Sr-$5s$ and Br-$4p$/Br-$4s$ are empty due to the electron configuration of $Sr^{2+}$ ([Kr]), and there is no potential for

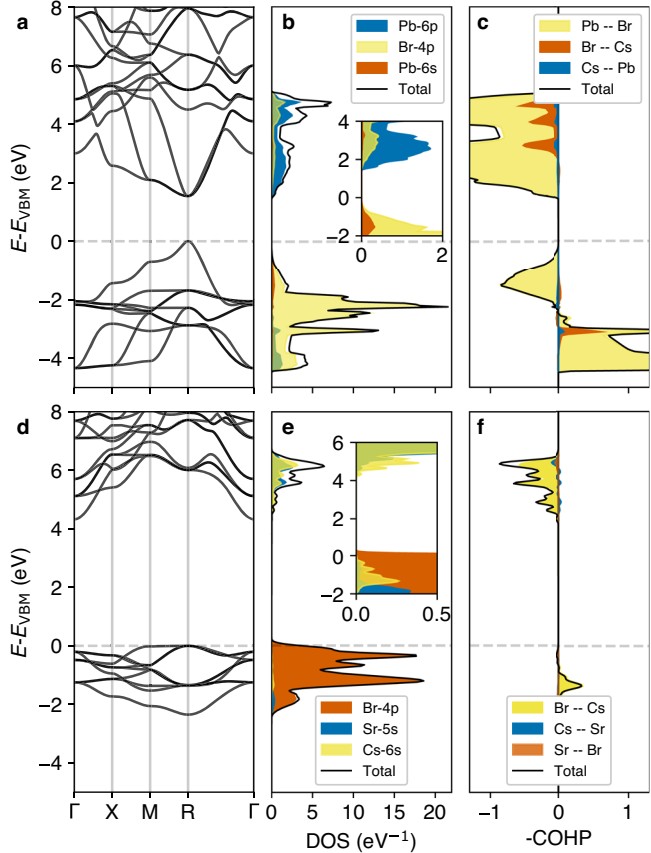

**Fig. 1 | Electronic structure.** DFT-computed electronic band structure of cubic $CsPbBr_3$ **a** and corresponding total and projected density of states (DOS, **b**) and crystal-orbital Hamilton population (COHP, **c**). Panels **d**–**f** show the same data for $CsSrBr_3$.

**Table 1 | Dielectric properties of cubic CsMBr₃**

| Compound | $\varepsilon_\infty$ | $Z^*_{Cs}$ | $Z^*_{M-site}$ | $Z^*_{Br}$ (xx, yy, zz) |
|---|---|---|---|---|
| CsPbBr₃ | 5.39 | 1.38 | 4.33 | (−0.63, −0.63, −4.46) |
| CsSrBr₃ | 3.02 | 1.35 | 2.43 | (−0.91, −0.91, −1.97) |

Dielectric constant in the high-frequency limit with respect to the optical phonon mode frequencies, $\varepsilon_\infty$, and Born effective charges, $Z^*_i$, of cubic CsPbBr₃ and CsSrBr₃ as calculated by DFT. We report $Z^*_{Br}$ for the Br bonded with Pb/Sr along the z axis.

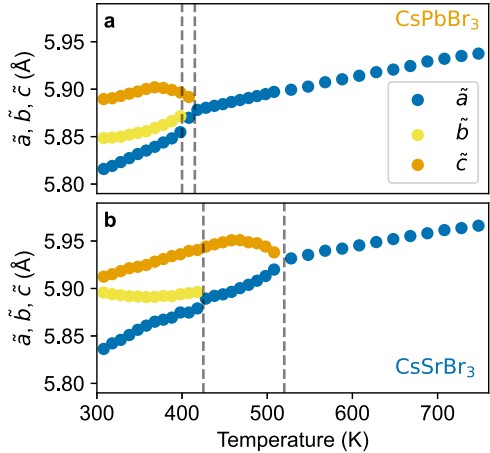

**Fig. 2 | Structural properties.** Temperature-dependent lattice parameters of CsPbBr₃ **a** and CsSrBr₃ **b** determined by XRD throughout the orthorhombic−tetragonal−cubic phases. We show reduced lattice parameters $\tilde{a}$, $\tilde{b}$ and $\tilde{c}$ for better visualization, with the orthorhombic phase expressed in the *Pbnm* setting. Dashed vertical lines indicate phase-transition temperatures. Error bars from Pawley fitting are smaller than the markers and are omitted.

lone pair formation on Sr²⁺. A manifestation of the lack of $ns^2$ cations in CsSrBr₃ is that there is no cross-gap hybridization of the halide valence orbitals. By contrast, Br-4*p* orbitals hybridize with Pb-6*p* across the gap of CsPbBr₃ (see the pCOHP in Fig. 1c). This leads to large Born effective charges, *i.e.*, large changes in the macroscopic polarization upon ionic displacements[48–51] reported in Table 1, which for CsPbBr₃ are more than double the formal charge of Pb (+2) and Br (-1) and much larger than the corresponding values for CsSrBr₃. Similarly, there is also a larger electronic contribution to the dielectric response in CsPbBr₃ and it features a larger value of the dielectric function at the high-frequency limit ($\varepsilon_\infty$) compared to CsSrBr₃.

## Structural properties and phase transitions

In spite of the markedly different electronic structure and bonding characteristics, CsSrBr₃ and CsPbBr₃ exhibit the same high-temperature cubic crystal structure ($Pm\bar{3}m$) and very similar lattice parameters (see Supplementary Note 3). One can rationalize this through the nearly identical ionic radii of Pb²⁺ and Sr²⁺ (119 and 118 pm) and the resulting Goldschmidt factors for the compounds (0.862 and 0.865). Furthermore, both materials exhibit the same sequence of structural phase transitions from the high-temperature cubic to the low-temperature orthorhombic phase (with an intermediate tetragonal phase), as shown by temperature-dependent lattice parameters in Fig. 2 that were determined via XRD. The cubic-to-tetragonal phase transition temperature of CsSrBr₃ (~520 K) is noticeably higher than that of CsPbBr₃ (~400 K)[52,53] and slightly higher (~10 K) than that reported for Eu-doped CsSrBr₃:Eu 5%[54]. The volumetric thermal expansion coefficient ($\alpha_V$) of CsSrBr₃ (~$1.32 \times 10^{-4}$ K⁻¹ at 300 K) is large and similar to that of CsPbBr₃ (~$1.29 \times 10^{-4}$ K⁻¹, see the Supplementary Note 3), in good agreement with the one reported for CsSrBr₃:Eu[54]. Just as for other inorganic HaPs, $\alpha_V$ of CsSrBr₃ slightly decreases with temperature[55,56]. The similarity of geometric factors and structural phase transitions suggests that the octahedral tilting dynamics in CsSrBr₃ might be similar to those in CsPbBr₃, which contrasts with their markedly different electronic structure, and prompts a deeper investigation of the impact of the $ns^2$ cations on structural dynamics.

## Lower-temperature lattice dynamics

We conduct IR and Raman spectroscopy at different temperatures as well as DFT-based harmonic-phonon calculations. The measured IR spectra show that the dominant CsSrBr₃ features are blue-shifted compared to those of CsPbBr₃ (see Fig. 3a). Indeed, our DFT calculations of IR activities find a significant softening of the infrared-active TO modes in CsPbBr₃ compared to those in CsSrBr₃ (see Fig. 3b): the most prominent IR-active TO mode in CsPbBr₃ and CsSrBr₃ appears at ~68 and 146 cm⁻¹, respectively, corresponding to the same irreducible representation ($B_{3u}$) with similar eigenvectors (see Supplementary Fig. 9) in each system. This is in line with the theory of weak PJT effects in general[29] and expectations for lone pairs in particular, with significant softening of ungerade modes in CsPbBr₃ that would correspond to lone-pair formation in the strong PJT case relative to those in CsSrBr₃. Notably, this softening is primarily driven by differences in bonding rather than the difference in the atomic masses (see Supplementary Note 7).

Moreover, the LO/TO splitting is enhanced in CsPbBr₃ compared to in CsSrBr₃ and the LO phonon modes are hardened. Related to this, the CsPbBr₃ IR spectrum exhibits a broad feature which is known as the Reststrahlen band as has been reported before for MA-based HaPs[57]. This particular effect results in near-zero transmission through the material in a frequency range between the TO and LO modes, represented by high IR intensity values, and occurs in polar materials with larger Born-effective charges. Because the TO modes are softened and the LO modes hardened in CsPbBr₃ compared to CsSrBr₃, and because the latter is less polar (*cf.* Table 1), the absence of the $ns^2$ cations leads to a much less pronounced, blue-shifted Reststrahlen band appearing in a smaller frequency window in CsSrBr₃ (see Fig. 3a, and Supplementary Note 4).

Figure 3c shows the 80 K Raman spectra of CsPbBr₃ and CsSrBr₃, which are in good agreement with the Raman activities calculated for harmonic phonons (Fig. 3d). Specifically, the experimental spectrum of CsPbBr₃ in Fig. 3b finds three broader features at frequencies below and one weaker-intensity feature at frequencies above 100 cm⁻¹. Conversely, CsSrBr₃ exhibits a structured feature around 50 cm⁻¹, a pronounced signal close to 100 cm⁻¹, and then a series of weaker intensities between 100 and 150 cm⁻¹. While the DFT-computed Raman activities calculated in the harmonic approximation are in broad agreement with these findings (see Fig. 3d), we note a slightly larger deviation of approximately 20 cm⁻¹ for the higher-frequency peak in CsPbBr₃. These findings lead us to conclude that unlike in IR, the Raman spectrum of CsSrBr₃ exhibits no substantial energy shifts with respect to CsPbBr₃. Computing the phonon DOS for the orthorhombic phase of both compounds with DFT (see Supplementary Fig. 8), we find that they exhibit similar phonon DOS below 100 cm⁻¹, *i.e.*, in the region of most of the Raman-active modes. The similar phonon DOS and the contributions of the M-site at low frequencies explain the limited shift of the CsSrBr₃ Raman spectrum, which might be surprising at first sight given the different atomic masses of Sr and Pb. Above this range, CsPbBr₃ exhibits few vibrational states while CsSrBr₃ shows its most pronounced phonon DOS peaks, which correspond well with the strongest IR mode calculated from the harmonic approximation.

## High-temperature lattice dynamics

A key signature of vibrational anharmonicity in HaPs at higher temperatures is the Raman central peak[26,33–35,43–45]. We use this feature that is nominally symmetry-forbidden in the cubic phase as a fingerprint to

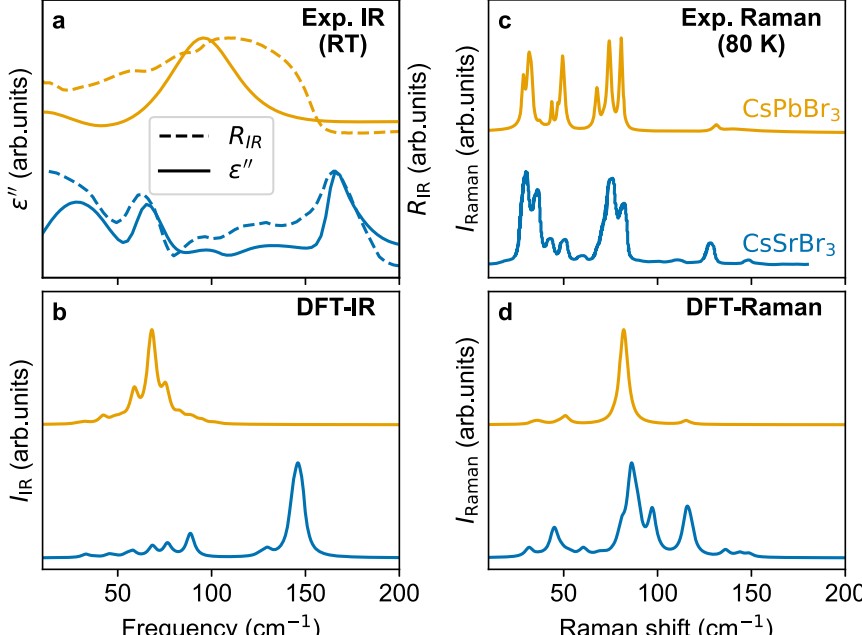

**Fig. 3 | Lattice dynamics at lower temperatures. a** IR-reflectivity spectra (dashed curves) and fitted imaginary part of the dielectric function (solid curves, see Supplementary Note 4 for details) of CsPbBr₃ and CsSrBr₃ measured at room temperature. **b** DFT-calculated IR-absorption spectra within the harmonic approximation for the orthorhombic phases. **c** Raman spectra of orthorhombic CsPbBr₃ and CsSrBr₃ measured at 80 K. **d** DFT-calculated Raman spectra of both compounds within the harmonic approximation for the orthorhombic phases.

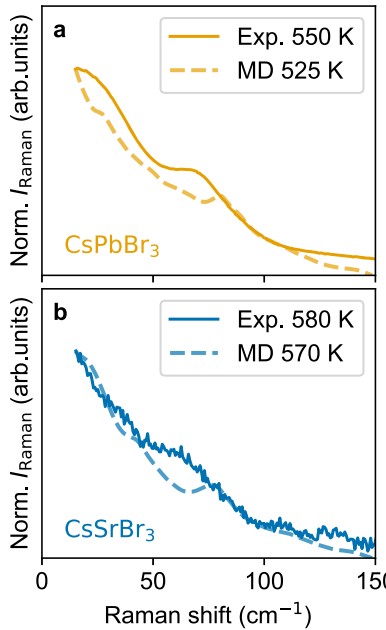

**Fig. 4 | Lattice dynamics at higher temperature.** Raman spectra of CsPbBr₃ **a** and CsSrBr₃ **b** in the high-temperature cubic phase measured experimentally and calculated using DFT-MD. The central peak appears for both compounds in the experiments and computations despite significant differences in bonding: $[PbBr_6]^{4-}$ is proximate to lone-pair formation (*i.e.*, exhibits a weak PJT effect)[29], while PJT effects associated with $[SrBr_6]^{4-}$ are negligible.

directly investigate how the propensity for cation lone-pair formation or lack thereof determines anharmonicity in these materials, using Raman spectroscopy and DFT-based MD simulations. Interestingly, a central peak also appears in the high-temperature Raman spectrum of CsSrBr₃ (see Fig. 4 and Supplementary Note 5 for full temperature range). We note that differences in Raman intensity imply that the

scattering cross-section of CsSrBr₃ is notably weaker than that of CsPbBr₃, which is due to its significantly higher bandgap and weaker dielectric response at the Raman excitation wavelength (785 nm) and because a powder sample of CsSrBr₃ has been used for which scattering of light in the back-scattering direction is considerably lower. The presence of a central peak in CsSrBr₃ shows that local fluctuations associated with a cation lone-pair are not required for the low-frequency diffusive Raman scattering and anharmonicity to occur. This result, together with the identical phase-transition sequences of both materials (see Fig. 2), led us to investigate the role of tilting instabilities in CsSrBr₃ and CsPbBr₃.

We first calculate the Raman spectrum for both compounds using MD calculations (see Fig. 4 and Methods section). Remarkably, a central peak appears also in the MD-computed high-temperature Raman spectrum of CsPbBr₃ and CsSrBr₃. We find good agreement between experiment and theory, both showing a feature between 50 and 100 cm⁻¹ in the Raman spectra of the two materials in addition to the central peak.

Next, we compute harmonic phonon dispersions of both compounds (see Fig. 5) and find these to be remarkably similar for cubic CsSrBr₃ and CsPbBr₃ in the low frequency region, in line with the aforementioned similarities in the phonon DOS of the orthorhombic phase. Specifically, both compounds exhibit the same dynamic tilting instabilities at the edge of the Brillouin zone (BZ), governed by in-phase (M point) and three degenerate out-of-phase (R point) rotations. These rotation modes are not only active in the phase transitions, but they also have been discussed to drive the dynamic disorder of halide perovskites[4,14,58–61].

Finally, using the MD trajectories of CsPbBr₃ and CsSrBr₃ in the cubic phase, we calculate the frequency-resolved dynamic changes of octahedral rotation angles, $\Phi_\alpha(\omega)$ (see Fig. 6 and Eq. (1) in the Methods Section). Figure 6b shows $\Phi_\alpha(\omega)$ for CsPbBr₃ and CsSrBr₃ and indicates strong low-frequency tilting components in both CsPbBr₃ and CsSrBr₃. Recently, a phenomenological model for the description of the temperature-dependent Raman spectra of cubic HaPs proposed the inclusion of a low-frequency anharmonic feature, which was associated

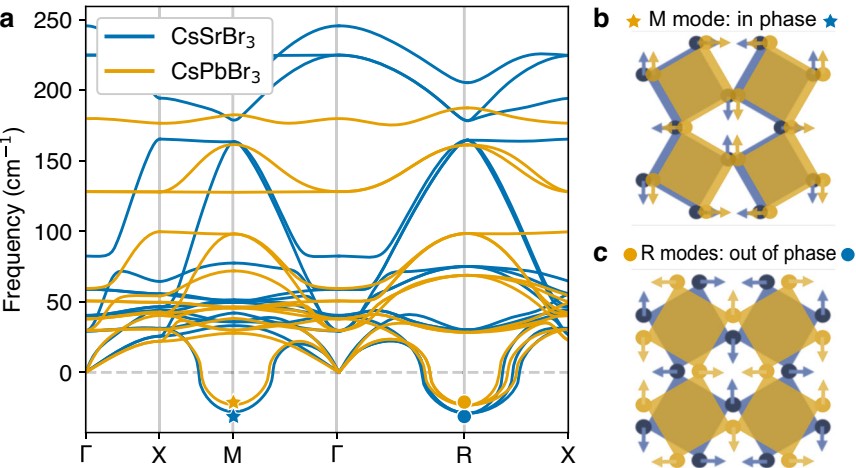

**Fig. 5 | Dynamic instabilities in the lattice dynamics. a** Harmonic phonon dispersion of cubic CsPbBr$_3$ and CsSrBr$_3$ showing the dynamic instabilities in the high-temperature, cubic phase of both compounds. The imaginary modes at the M and R points are the in-phase and out-of-phase tilting depicted in **b, c**, respectively. The tilting modes are almost identical for CsSrBr$_3$ and CsPbBr$_3$.

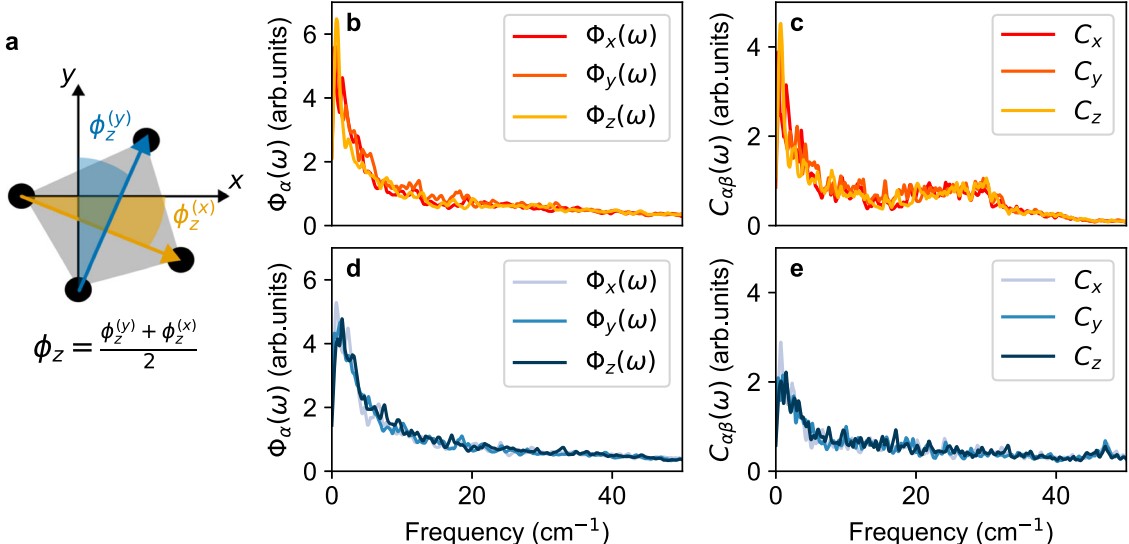

**Fig. 6 | Impact of cation electron configuration on octahedral dynamics at higher temperature. a** Schematic representation of the MBr$_6$ octahedron aligned along the $z$ Cartesian axis. The octahedral rotation angle around $z$, $\phi_z$, is defined as the average of the angles formed by the $x/y$ Cartesian axis and the vector connecting two in-plane Br atoms at opposing edges of the octahedron ($\phi_z^{(x)}$ in red and $\phi_z^{(y)}$ in blue). Note that a clockwise rotation is defined as positive and counterclockwise as negative. Fourier transform of the octahedral rotation angle, $\Phi_\alpha(\omega)$, and cross-correlation between rotation angle and M-site displacement, $C_{\alpha\beta}(\omega)$, calculated using DFT-MD trajectories of cubic CsPbBr$_3$ (**b, c**, respectively, $T = 525$ K) and CsSrBr$_3$ (**d, e**, respectively, $T = 570$ K).

with transitions between minima of a double-well potential energy surfaces[45] that correspond to different octahedral tiltings[60,62–64]. Our results confirm that substantial octahedral dynamics correspond to low-frequency features dynamically breaking the cubic symmetry in CsPbBr$_3$ and CsSrBr$_3$[4,14,44,65,66]. Interestingly, this low-frequency component appears irrespective of the presence of $ns^2$ cations and induces the formation of relatively long-lived (tens of ps) structural distortions (see Supplementary Fig. 11), which strongly deviate from the average cubic symmetry. This suggests that the dynamic deviations from the long-range, crystallographic structure enable the low-frequency Raman response without violating the selection rules.

We investigate the impact of the M-site chemistry on octahedral tilting tendencies[33] by computing the Fourier-transform of cross-correlations between rotation angles and M-site displacements, $C_{\alpha\beta}(\omega)$ (see Eq. (2) in the Methods section). Larger values of $C_{\alpha\beta}$ generally indicate stronger coupling between octahedral rotations and Pb displacements. Absence of the propensity for lone-pair formation becomes evident in the low intensity of $C_{\alpha\beta}(\omega)$ for CsSrBr$_3$ (Fig. 6c), which is less than half of that of CsPbBr$_3$, especially at low-frequencies relevant for the slow, anharmonic, symmetry-breaking rotational features. This suggests that the presence of the $ns^2$ cations in CsPbBr$_3$ enhances the low-frequency octahedral tilting, in line with the literature[33]. M-site displacements and octahedral rotations are correlated because the latter is accompanied by changes of the Br-Pb-Br resonant network[17] affecting the charge density in the vicinity of the M-site. While this effect is very weak in CsSrBr$_3$ (see Supplementary Fig. 12), the non-zero $C_{\alpha\beta}$ for this case shows that the presence of $ns^2$ cations is not necessary to couple octahedral rotations and M-site displacements because the ions are still interacting through other types of interactions, *e.g.*, electrostatically or due to Pauli repulsion. In CsPbF$_3$, the interaction of tilting and M-site displacements is strong enough to drive the adoption of an unusual tilt pattern[37]. We speculate

that the lone-pair-enhanced tilting could contribute to the fact that $CsPbBr_3$ has a lower tetragonal-to-cubic phase transition temperature compared to that of $CsSrBr_3$.

## Discussion

We directly disentangled structural and chemical effects in HaPs by comparing $CsPbBr_3$ and $CsSrBr_3$, two compounds with similar ionic radii and structural properties but entirely different orbital interactions that imbue $CsPbBr_3$ with the weak PJT effect common to technologically-relevant Pb perovskites and $CsSrBr_3$ with negligible PJT effects. While the $ns^2$ configuration of the octahedral cations is paramount for the optoelectronic and dielectric properties of these materials, using the Raman central peak at higher temperatures as a fingerprint to detect anharmonicity we found it to appear also for $CsSrBr_3$ with $5s^0$ cations and to correlate with slow, anharmonic rotations of the octahedra. Notably, the anharmonicity of the tilting motions is different from that of intra-octahedral distortions associated with the PJT effect[29]. Altogether, these findings demonstrate that the perovskite structure allows for anharmonic vibrational dynamics to occur, irrespective of the presence of $ns^2$ cations with the propensity to form lone pairs, which establishes this somewhat unusual behavior as a generic effect in this material class. We note that recent work by some of the present authors has investigated the commonalities and differences between oxide perovskites and HaPs in this context[45].

Since octahedral dynamics impact the optoelectronic characteristics of these systems, our results have implications for synthesis of new HaPs with improved properties for technological applications. For instance, Pb-Sr alloying has been proposed as a method to tune the band gap of HaPs for light emission and absorption applications[47]. Our work implies that such Sr alloying for tuning electronic and dielectric properties preserves the strongly anharmonic lattice dynamics. Furthermore, investigating related compounds with distinct electronic configurations on the octahedral cation, such as $CsEuBr_3$, may provide further insight about chemical trends in tuning of the HaP properties.

The relevance of these findings for material design strategies of HaP compounds is additionally affirmed when putting our results in the context of previous work discussing anharmonic effects in this class of materials. Specifically, cubic $CsPbBr_3$, $CsSnBr_3$, $CsGeBr_3$, $(CH_3NH_3)_{0.13}(CH_3CH_2NH_3)_{0.87}PbBr_3$, $CH(NH_2)_2PbBr_3$, and, here, $CsSrBr_3$ are all reported to exhibit dynamic hopping between low symmetry minima on the potential energy surface[33–35,67]. By contrast, the high symmetry phase of $Cs_2AgBiBr_6$ is anharmonically stabilized and exhibits well-defined normal modes and a soft-mode transition on cooling[42]. $Cs_2SnBr_6$, on the other hand, lacks any phase transitions and similarly exhibits well-defined normal modes[68]. Where previously the strength of the PJT effect associated with $ns^2$ cations or the density of such cations appeared to be a plausible predictor of broad, nominally symmetry-forbidden Raman scattering resulting in a central peak, our work suggests that instead the differing symmetry in both the structure and the chemical bonding of metal halide perovskites and double-perovskites may be a controlling factor. Notably, $CsGeBr_3$, which exhibits no octahedral tilting transitions[69] and a broad Raman central peak in the cubic phase with a mode reflecting persistent pyramidal $[GeBr_3]^-$ anions[33], corresponds to the strong PJT[29] case: Stereochemically expressed cation lone pairs are evident in the low temperature average structure[69] and in the local fluctuations of the cubic phase[33]. Dynamic symmetry-breaking giving rise to a broad Raman central peak is thus observed for three distinct bonding regimes with regard to pseudo-Jahn–Teller effects: strong PJT ($CsGeBr_3$)[33], weak PJT ($CsPbBr_3$ and others)[34], and negligible PJT ($CsSrBr_3$).

In conclusion, the $ns^2$ electron configuration in HaPs that can result in the formation of lone-pairs is crucial to several favorable electronic features[26,39,46] and gives rise to the elevated ionic dielectric response via enhancement of Born effective charges[39,48]. However, we found that presence of a strong or weak PJT effect associated with $ns^2$ cations is not necessary to produce dynamic symmetry-breaking of the sort that gives rise to broad, intense Raman scattering in the high temperature phases of HaPs and that has been associated with the unique optoelectronic properties in these compounds such as long charge-carrier lifetimes and photoinstabilities. Instead, such dynamic symmetry breaking is common to all cubic bromide and iodide (single-)perovskites thus far studied to the best of our knowledge. These results highlight the key role of structural chemistry in the anharmonic dynamics of halide perovskites, providing an additional criterion for the design of soft optoelectronic semiconductors.

## Methods

### Electronic structure calculations

DFT calculations were performed with Vienna ab-initio simulation package (VASP) code[70] using the projector augmented wave (PAW) method[71]. We employed the Perdew-Burke-Ernzerhof (PBE) exchange-correlation functional[72] and the Tkatchenko-Scheffler (TS) scheme[73]—using an iterative Hirshfeld partitioning of the charge density[74,75] – to account for dispersive interactions. This setup has been shown to accurately describe the structure of HaPs, also in regard to the omission of SOC which impacts electronic-structure properties but does not result in significant changes of quantities related to the total energy[76,77]. All static calculations used an energy convergence threshold of $10^{-6}$ eV, a plane-wave cutoff of 500 eV, and a $\Gamma$-centered $k$-grid of $6 \times 6 \times 6$ ($6 \times 4 \times 6$) for the $Pm\bar{3}m$ ($Pnma$) structures. Lattice parameters were optimized by a fitting procedure using the Birch-Murnaghan equation of state[78,79] The final structures used in all subsequent calculations were obtained by relaxing the ionic degrees of freedom until the maximum residual force was below $10^{-4}$ eV Å$^{-1}$. The total and projected electronic DOS and COHP, were calculated by partitioning the DFT-calculated band structure into bonding and antibonding contributions using the LOBSTER code[80,81]. For this task, the DFT-computed electronic wave functions were projected onto Slater-type orbitals (basis set name: pbeVaspFit2015)[80] including Cs 6s, 5p and 5s, Pb 6s and 6p, and Br 4p and 4s states. The maximum charge spilling in this procedure was 1.3%. Spin-orbit coupling was not included in our calculations, since it is currently not supported by the LOBSTER code. We emphasize that our focus is on the orbital contributions to the (anti) bonding interactions, rather than on a quantitative descriptions of the energy.

### Phonon calculations

Phonon dispersions and DOSs were obtained via the finite displacements method implemented in the phonopy package[82]. For these calculations, we used $2 \times 2 \times 2$ supercells with 40 (160) atoms of the $Pm\bar{3}m$ ($Pnma$) $CsMBr_3$ structures reducing $k$-space sampling accordingly. IR and Raman spectra were computed with the phonopy-spectroscopy package[83], using zone-center phonon modes, Born-effective charges and polarizabilities, calculated with density functional perturbation theory (DFPT)[84].

### First-principles molecular dynamics

DFT-based MD calculations were performed for $2 \times 2 \times 2$ supercells of the $Pm\bar{3}m$ structures using a Nosé-Hoover thermostat within the canonical ensemble (NVT), as implemented in VASP[85]. The simulation temperature was set to $T$=525 and 570 K for $CsPbBr_3$ and $CsSrBr_3$, respectively. An 8 fs timestep, reduced $k$-grid of $3 \times 3 \times 3$, and energy convergence threshold of $10^{-5}$ eV were used for the 10 ps equilibration and 115 ps production runs.

### Raman spectra from molecular dynamics

DFT-based MD calculations were used to compute the high-temperature Raman spectra of $CsPbBr_3$ and $CsSrBr_3$. We calculated Raman intensities from the autocorrelation function of the

polarisability, as detailed elsewhere[86]. The polarizabilities were calculated with DFPT[84] on 400 evenly-spaced snapshots every 0.11 ps for a total of 44.8 ps. The $k$-grid employed for the DFPT calculations was set to $4 \times 4 \times 4$ after testing convergence of the polarisability tensor for several snapshots.

## Octahedral rotation dynamics and cross-correlations

We quantified the octahedral dynamics using the rotation angles, $\phi_\alpha$, around a given Cartesian axis $\alpha$ (see Fig. 6a). The frequency-resolved rotational dynamics were calculated as the Fourier transform of $\phi_\alpha$:

$$\Phi_\alpha(\omega) = \frac{1}{N_{\text{steps}}} \int_0^\infty \phi_\alpha(t) e^{-i\omega t} dt, \quad (1)$$

where $N_{\text{steps}}$ is the number of snapshots. To compute the angles we selected 1000 equally spaced snapshots. We calculated the frequency-resolved cross-correlation between octahedral rotation angles (around a Cartesian direction $\alpha$) and the displacements (along a Cartesian direction $\beta$) of the corresponding M-site, $d_\beta^M(t)$, as:

$$C_{\alpha\beta}(\omega) = \frac{1}{N_{\text{steps}}} \int_0^\infty \frac{\langle \phi_\alpha(t+\delta t) \cdot d_\beta^M(t) \rangle}{\langle \phi_\alpha(t) \cdot d_\beta^M(t) \rangle} e^{-i\omega t} dt. \quad (2)$$

## Polycrystalline sample preparation

CsBr (Alfa Aesar, 99.9%), anhydrous SrBr$_2$ (Alfa Aesar, 99%), Cs$_2$CO$_3$, PbO, and concentrated aqueous HBr were purchased and used as received. Guided by the reported pseudo-binary phase diagram[87], polycrystalline CsSrBr$_3$ for X-ray powder diffraction and Raman spectroscopy was prepared by a solid-state reaction at 600 °C. CsBr (5 mmol, 1064 mg) and SrBr$_2$ (5 mmol, 1237 mg) were ground and pressed into a 5 mm diameter pellet, placed in an alumina crucible, and flame-sealed under ∼1/3 atmosphere of argon in a fused silica ampoule. The reaction yields a porous, colorless pellet which is easily separated from the crucible and ground in inert atmosphere. Polycrystalline CsPbBr$_3$ for X-ray powder diffraction was prepared in ambient atmosphere by precipitation from aqueous hydrobromic acid. PbO (2 mmol, 446.4 mg) was dissolved in 2 mL hot concentrated HBr under stirring. Cs$_2$CO$_3$ (1 mmol, 325.8 mg) was added slowly resulting in an immediate bright orange precipitate. 13 mL additional HBr was added and the mixture left to stir. After an hour, stirring was stopped and the mixture allowed to cool to room temperature. Excess solution was decanted, and the remaining mixture was evaporated to dryness on a hotplate and ground. Phase purity of all prepared compounds was established by powder XRD.

## Single crystal preparation

Single crystals of CsSrBr$_3$ were grown by the Bridgman method from a stoichiometric mixture of the binary metal bromides in a 10 mm diameter quartz ampoule. CsSrBr$_3$ was pulled at 0.5 mm h$^{-1}$ through an 800 °C hot zone, yielding a multi-crystalline rod from which several-mm single crystal regions could be cleaved. CsSrBr$_3$ is hygroscopic and all preparation and handling was performed in an inert atmosphere.

The vertical Bridgman method was used to grow large, high-quality single crystals of CsPbBr$_3$. After synthesis and purification (see Supplementary Note 2 for details), the ampoule was reset to the hot zone for the Bridgman Growth. The zone 1 temperature was set to 650 °C with a 150 °C h$^{-1}$ ramp rate, and held for 12 h to ensure a full melt before sample motion occurred. The zone 2 and 3 temperatures were set to 375 °C. These temperatures were held for 350 h while the ampoule was moved through the furnace at a rate of 0.9 mm h$^{-1}$ under 0.3 rpm rotation. After the motion had ceased, the zone 1 temperature ramped to 375 °C to make the temperature profile in the furnace uniform. The cooling program was set to slow during the phase transitions occurring near 120 and 90 °C, with a 10 °C h$^{-1}$ cooling rate from 375 °C

to 175 °C, a 2.5 °C h$^{-1}$ slow cooling rate from 175 °C to 75 °C, and a 10 °C h$^{-1}$ rate to 30 °C. The resulting CsPbBr$_3$ ingot was orange-red and had large (≥5 mm) transparent single-crystalline domains, though the edges of some portions exhibited twinning.

## X-ray diffraction

Polycrystalline samples were ground with silicon powder (as an internal standard and diluent) and packed in borosilicate glass capillaries. Powder XRD patterns were measured in STOE geometry using a STOE Stadi P diffractometer (Mo K$_{\alpha 1}$ radiation, Ge-(111) monochromator, Mythen 1K Detector) equipped with a furnace. Data were analyzed by sequential Pawley refinement using GSAS-II[88].

## Infrared reflectivity measurements

IR-reflection spectra in the THz range were measured as a combination of THz spectroscopy (TDS) for the low-frequency end and bolometer detection for the higher frequencies. Bolometer spectra were measured using a Bruker 80v Fourier-transform IR spectrometer with a globar source and a bolometer detector cooled to liquid He temperatures. The crystals were mounted for reflection measurements and the instrument was sealed in vacuum. A gold mirror was used as reflection reference. TDS was performed using a Spectra Physics Mai Tai-Empower-Spitfire Pro Ti:Sapphire regenerative amplifier. The amplifier generates 35 fs pulses centered at 800 nm at a repetition rate of 5 kHz. THz pulses were generated by a spintronic emitter, which was composed of 1.8 nm of Co$_{40}$Fe$_{40}$B$_{20}$ sandwiched between 2 nm of tungsten and 2 nm of platinum, all supported by a quartz substrate. The THz pulses were detected using electro-optic sampling in a (100)-ZnTe crystal. A gold mirror was used as reflection reference. The sample crystals, THz emitter and THz detector were held under vacuum during the measurements.

TDS offers better signal at low frequency, while bolometer measurements have an advantage over TDS at higher frequencies. Therefore, the spectra were combined and merged at 100 cm$^{-1}$. Owing to scattering losses, the absolute intensity of reflected light can not be taken quantitatively. Therefore, the spectra were scaled to the signal level at 100 cm$^{-1}$ before merging the data. The final reflectivity spectra are given in arbitrary units. The phonon frequencies and overall spectral shape allows for fitting to the dielectric function.

## Raman spectroscopy

All the measurements were taken in a home-built back scattering Raman system. For all measurements, the laser was focused with a 50× objective (Zeiss, USA), and the Rayleigh scattering was then filtered with a notch filter (Ondax Inc., USA). The beam was focused into a spectrometer 1 m long (FHR 1000, Horiba) and then on a CCD detector. To get the unpolarized Raman spectrum for the single crystals (CsSrBr$_3$ and CsPbBr$_3$), two orthogonal angles were measured in parallel and cross configurations (four measurements overall). The unpolarized spectrum is a summation of all four spectra. The samples were cooled below room temperature by a Janis cryostat ST-500 controlled by Lakeshore model 335 and were heated above room temperature by a closed heating system (Linkam Scientific). Due to the sensitivity of CsSrBr$_3$ to ambient moisture, CsSrBr$_3$ powder was flame-sealed in a small quartz capillary for the high-temperature measurements, and a single crystal was loaded into a closed cell under an Ar environment for the low temperatures measurements. CsSrBr$_3$ low temperature measurements were taken with a 2.5 eV CW diode laser (Toptica Inc.). CsSrBr$_3$ high-temperature measurement and all the CsPbBr$_3$ measurements were taken with a 1.57 eV CW diode laser (Toptica Inc.). We note that while Raman spectra on quartz show a contribution towards zero frequency[89], it is narrower in frequency than what we observe. Results from control experiments (see

Supplementary Fig. 7) show that the main signals from quartz do not contribute to the measured Raman spectra of $CsSrBr_3$.

## Data availability

The computational and experimental data generated in this study have been deposited in the *Zenodo* repository under accession code 10975217[90].

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

## Acknowledgements

Funding provided by the Alexander von Humboldt-Foundation in the framework of the Sofja Kovalevskaja Award, endowed by the German Federal Ministry of Education and Research, by the Deutsche Forschungsgemeinschaft (DFG, German Research Foundation) via Germany's Excellence Strategy - EXC 2089/1-390776260, and by TU Munich - IAS, funded by the German Excellence Initiative and the European Union Seventh Framework Programme under Grant Agreement No. 291763, are gratefully acknowledged. Funding was provided by the Engineering and Physical Sciences Research Council (EPSRC), UK. The Gauss Centre for Supercomputing eV is acknowledged for providing computing time through the John von Neumann Institute for Computing on the GCS Supercomputer JUWELS at Jülich Supercomputing Centre. D.H.F. gratefully acknowledges financial support from the Alexander von Humboldt Foundation and the Max Planck Society. D.H.F. thanks Maximilian A. Plass for assistance with flame-sealing the Raman capillaries. K.M.M. and M.V.K. acknowledge financial support by ETH Zürich through the ETH+ Project SynMatLab Laboratory for Multiscale Materials Synthesis.

## Author contributions

S.C.-D. performed the theoretical calculations, analyzed the data and wrote the initial manuscript under the supervision of D.A.E and with additional support by M.G. A.C. performed Raman measurements and analyzed data under the supervision of O.Y. S.M. performed IR measurements and analyzed data under the supervision of L.M.H. D.H.F. prepared polycrystalline samples, performed XRD measurements, analyzed data, and modeled the PJT effect. M.I. prepared the CsSrBr$_3$ single crystals. K.M.M. prepared the CsPbBr$_3$ single crystals under the supervision of M.V.K. D.H.F. and D.A.E. conceived and supervised the project.

## Funding

## Competing interests

The authors declare no competing interests.
