## [Peer Review File · Nature Communications]

Disentangling the Effects of Structure and Lone-Pair Electrons in the Lattice Dynamics of Halide PerovskitesReviewers' comments:

Reviewer #1 (Remarks to the Author):

The manuscript by Caicedo-Dávila et al. delves into an investigation of whether the presence of a lone pair electron (LPE) is a crucial factor in determining the strongly anharmonic lattice dynamics, particularly characterized by the Raman central peaks, in halide perovskite materials. The authors conducted a comprehensive analysis encompassing spectroscopic characterizations and theoretical simulations on two structurally similar compounds, CsPbBr₃ and CsSrBr₃. Their findings revealed strikingly similar lattice dynamics between the two materials. Notably, the Raman spectrum of the high-temperature cubic CsSrBr₃ displayed a broad central peak, akin to that observed in CsPbBr₃. This was particularly intriguing as CsSrBr₃ lacks an LPE, and both structures exhibited closely analogous phase transitions linked to octahedral tilting instabilities. Consequently, the authors concluded that dynamic octahedral tilting is the driving force behind the Raman central peaks in both structures, and the presence of an LPE is not a prerequisite for these peaks in halide perovskites. Overall, the study's scientific methodologies are robust, and the results offer valuable insights into the structural properties and lattice dynamics of halide perovskites, making it a very strong candidate for publication in Nature Communications. However, there are several minor points that require clarification before publication

1. It remains unclear whether the authors determined the crystal structure of CsSrBr₃. As far as I know, this structure does not appear in the crystallographic database. It would be beneficial to provide temperature-dependent PXRD data for the samples and provide the refined crystal structure used in the simulations.
2. The similarity in Raman spectra between the two structures, despite the significant difference in atomic masses between Sr and Pb, is surprising to me. While the authors have proposed theoretical explanations, it is advisable to investigate CsEuBr₃, which lacks an LPE and adopts the perovskite structure, to further validate their theory.
3. In Figure 4, there appears to be a noticeable difference in Raman intensity between CsSrBr₃ and CsPbBr₃, as indicated by the signal-to-noise ratio. Could the authors clarify this observation and provide a comparison of Raman intensities between the two structures in their theoretical simulations?
4. The authors have suggested that the presence of an LPE in CsPbBr₃ enhances low-frequency octahedral tilting. However, it remains unclear why octahedral tilting and B-cation displacement are correlated, given that these two types of distortions are typically mutually repulsive from a geometrical perspective.
5. Previous studies have demonstrated an anharmonic and even a double-well potential energy surface for B-cation displacement in cubic CsPbBr₃ upon thermal heating, *J. Am. Chem. Soc.* 2016, 138, 11820. This is expected to contribute to lattice anharmonicity. Could the authors clarify whether and how this factor may contribute to the differences between the two structures?
6. I just found that quartz crystal exhibits low-frequency Raman intensity at high temperatures, see *J. Chem. Phys.* 56, 1528–1533, 1972. A control experiment is recommended. Additionally, in Figure S3, the baselines on the anti-Stokes side of the high-temperature spectra appear higher than those on the Stokes side. What is the reason for this discrepancy?
7. Central peaks have been observed in other materials, particularly oxide perovskites. It would be valuable to compare the findings in halide perovskites with those observed in other systems to gain a broader perspective on the phenomenon.

Reviewer #2 (Remarks to the Author):

The manuscript “Disentangling the Effects of Structure and Lone-Pair Electrons in the Lattice Dynamics of Halide Perovskites” is an interesting look at the role of the metal’s electron configuration in the lattice dynamics of a halide perovskite. The experimental work is well done, as is the theory. My main critique is the framing of the article—the similar behavior between metals with noble gas and $d^{10}s^2$ electron configuration is not surprising to this reviewer, and indeed is what should be expected from the theory surrounding Jahn-Teller and pseudo-Jahn-Teller distortions.

The $(\text{SrBr}_6)^{4-}$ cluster has the same electron configuration as the $(\text{TiO}_6)^{8-}$ cluster, which is present in many oxide perovskites such as SrTiO_3 and BaTiO_3 . Ti^{4+} has a noble gas electron configuration, just like Sr^{2+} , so the behavior of the metals should be very similar. $(\text{PbBr}_6)^{4-}$ undergoes similar pseudo-Jahn-Teller distortions, which the authors of this manuscript know well (e.g., (1) Fabini, D. H.; Seshadri, R.; Kanatzidis, M. G. *MRS Bulletin* 2020, 45 (6), 467–477. <https://doi.org/10.1557/mrs.2020.142>). I suggest Bersuker, I. B. *Chemical Reviews* 2013, 113 (3), 1351–1390. <https://doi.org/10.1021/cr300279n> as a good reference, specifically Table 6.

The framing of the anharmonic effects in APbX_3 halide perovskites as a function of the lone pair is likely the cause of this confusion. The term “stereochemically active lone pair” is common in the literature, but in my mind is misleading because the lone pair itself is not the cause of the distortions—they originate from degeneracies in the first excited state/the pseudo-Jahn-Teller effect, which is why similar distortions occur in CsSrBr_3 , just like they do in ATiO_3 oxide perovskites. This confusing nomenclature is discussed in a few places in the recent halide perovskite literature—for instance, in Straus, D. B.; Mitchell Warden, H. E.; Cava, R. J. *Inorganic Chemistry* 2021, 60 (17), 12676–12680. <https://doi.org/10.1021/acs.inorgchem.1c01277>.

This manuscript should be published in *Nature Communications* after being revised to highlight the similar role of the pseudo-Jahn-Teller effect in both APbX_3 and ASrX_3 halide perovskites as well as emphasizing the similar electron configurations of the titanite oxide perovskites with CsSrBr_3 .

Reviewer #3 (Remarks to the Author):

This paper investigates the relationship between lattice anharmonicity and presence of lone-pair electrons (LPE) in metal halide perovskites (HaPs) by combining first-principal simulations with IR and Raman spectroscopy measurements on CsPbBr_3 and CsSrBr_3 .

The central result that the authors establish is that “the presence of a lone-pair electrons (LPE) is not required for the strong anharmonicity in the low-energy lattice dynamics of soft HaP semiconductors.” This result is established mainly based on computational modeling, partially validated with Raman spectroscopy. This result confirms prior conclusions previously reported in the literature (namely the fact that LPE is not required for strong anharmonicity in HaPs), which is appropriately cited by the authors. While the authors here confirm this conclusion by means of a comparison between CsPbBr_3 (LPE) and CsSrBr_3 (no LPE), I do not find that this provides sufficiently new insights to warrant the high visibility of a

publication in Nature Communications.

The two compositions studied show great similarity in crystal structure and phase transition sequences, and Sr lacks active lone-pair electrons (LPE) compared to Pb. The observation of a similar central peak in Raman spectra for both compositions indicates that LPE are not necessary for anharmonic octahedral tilting. In general, the information from the paper is quite simple and straightforward, but the results are not surprising or really novel.

Overall, the main point that LPE is not necessary for the presence of low-frequency anharmonic octahedral rotations is solid, based on simulations and the observation of a central Raman peak in cubic CsSrBr₃. However, I disagree that the simulations in the orthorhombic phase show "good agreement" with the measurements. Perhaps the MD simulations need to be better benchmarked by comparing with previously reported experimental data or other simulations.

The paper could also benefit from a deeper investigation of the simulations, to clarify how LPE could interact with anharmonic low-frequency modes, and thus result in stronger Cab(w) in CsPbBr₃ than CsSrBr₃.

The discussion feels diluted, reiterating points already made in intro and results.

Overall, I find that the results in the manuscript are a bit too thin, and the impact too limited, to warrant publishing in Nature Communications. I recommend shortening the paper and publishing in a more specialized journal.

More detailed comments and questions:

1. In Fig 3 the authors show a comparison between IR / Raman measurements and DFT simulations. The effects of LPE on the softening of TO mode was discussed to rationalize the difference between CsSrBr₃ and CsPbBr₃, but the discrepancies between simulations and experiments (both IR and Raman spectra) are rather drastic. The measured IR spectrum is much broader than simulations, potentially because the measurements are done at RT while Raman were done at 80K, closer to the orthorhombic phase DFT simulations. I am not sure why RT IR data are shown in the section titled 'lower-temperature lattice dynamics'. I think the overall message the authors want to deliver in this part is not quite clear. The large discrepancy between simulations and experiments needs to be acknowledged, and should be investigated and explained more fully.
2. The authors performed DFT-MD simulations to investigate the dynamics of octahedral rotations and concluded that the strong central peak in measured Raman spectra originates from low-frequency anharmonic modes and that LPE is not necessary for these anharmonic modes to occur. Perhaps the authors could also calculate the Raman spectrum based on their MD trajectories and make some comparisons between the high temperature Raman data to further benchmark their simulations.
3. The Raman spectrum of CsSrBr₃ shown in Fig. S3 indicates that the system still has a strong central peak in the tetragonal phase below the cubic-tetragonal phase transition, while the central peak is gone in orthorhombic phase, with a strong peak appearing at ~30cm⁻¹ instead. Could the authors explain what could be happening to account for this difference between the tetragonal and orthorhombic phases?

4. The authors note that “The cubic-to-tetragonal phase transition temperature of CsSrBr₃ (~520K) is noticeably higher than that of CsPbBr₃ (~400K)” but do not at all explain why. Please explain (preferably beyond just speculating).

5. The authors should label relevant information on the plots of Fig 3. Indicate where are the Reststrahlen band, the LO and the TO modes. Also, clarify how one is supposed to see “good agreement” between measured and computed spectra, for instance by matching peak positions and intensities (I tried but could not see “good agreement”).

Reviewer 1:

Comment 1: The manuscript by Caicedo-Dávila et al. delves into an investigation of whether the presence of a lone pair electron (LPE) is a crucial factor in determining the strongly anharmonic lattice dynamics, particularly characterized by the Raman central peaks, in halide perovskite materials. The authors conducted a comprehensive analysis encompassing spectroscopic characterizations and theoretical simulations on two structurally similar compounds, CsPbBr₃ and CsSrBr₃. Their findings revealed strikingly similar lattice dynamics between the two materials. Notably, the Raman spectrum of the high-temperature cubic CsSrBr₃ displayed a broad central peak, akin to that observed in CsPbBr₃. This was particularly intriguing as CsSrBr₃ lacks an LPE, and both structures exhibited closely analogous phase transitions linked to octahedral tilting instabilities. Consequently, the authors concluded that dynamic octahedral tilting is the driving force behind the Raman central peaks in both structures, and the presence of an LPE is not a prerequisite for these peaks in halide perovskites. Overall, the study's scientific methodologies are robust, and the results offer valuable insights into the structural properties and lattice dynamics of halide perovskites, making it a very strong candidate for publication in Nature Communications. However, there are several minor points that require clarification before publication

Response: We thank the reviewer for their strong support of our work for Nature Communications (“... making it a very strong candidate for publication in Nature Communications”) and providing comments to clarify certain aspects of our work. We address all points below.

Comment 2:

It remains unclear whether the authors determined the crystal structure of CsSrBr₃. As far as I know, this structure does not appear in the crystallographic database. It would be beneficial to provide temperature-dependent PXRD data for the samples and provide the refined crystal structure used in the simulations.

Response:

Indeed, the structures of CsSrBr₃ are not present in the Inorganic Crystal Structure Database, but Loyd and coworkers [J. Cryst. Growth 481 (2018) 35–39] report that lightly doped samples of CsSrBr₃:Eu adopt the common GdFeO₃-type orthorhombic perovskite structure with $a^-b^+a^-$ tilting. This structure indexes our diffraction data and is found to be mechanically stable by harmonic phonon calculations. The emphasis in our high temperature powder diffraction experiments was on thermal expansion and comparing the lattice parameters to those of CsPbBr₃. Therefore, our data were collected to the quality necessary for whole-pattern (Pawley) fitting but not to the quality necessary for Rietveld refinement of reliable atom positions and atomic displacement parameters. We have added a comparison of our refined lattice parameters to those of Loyd and coworkers for CsSrBr₃:Eu to the revised Supplementary Material, showing excellent agreement at low and intermediate temperatures, and a difference of roughly one part in 10⁴ at high temperatures. Since this information is not available in the literature, we added the DFT-optimized orthorhombic structure to the Supplementary Material as well.

Changes: The following figure has been added to the Supplemental Material, p. 2:

Figure S2: **Comparison of temperature-dependent lattice parameters.** Comparison of reduced lattice parameters from powder diffraction for CsSrBr₃ (this work) and CsSrBr₃:Eu,[S3] with the orthorhombic phase expressed in the *Pnma* setting. Agreement is favorable, with a possible small discrepancy in transition temperatures and cubic phase lattice parameter.

The following table has been added to the Supplemental Material, p. 3:

Table S1: Fractional coordinates of DFT-optimized CsSrBr₃ in the *Pnma* phase with lattice constants (8.281, 11.797, 8.234). The coordinates are reported per element proceeding along Cs (4 atoms), Sr (4 atoms), and Br (12 atoms).

0.0782794124058532	0.7500000000000000	0.0237544762301769
0.9217205875941468	0.2500000000000000	0.9762455237698231
0.4217205875941468	0.2500000000000000	0.5237544762301769
0.5782794124058532	0.7500000000000000	0.4762455237698231
0.5000000000000000	0.5000000000000000	0.0000000000000000
0.0000000000000000	0.5000000000000000	0.5000000000000000
0.5000000000000000	0.0000000000000000	0.0000000000000000
0.0000000000000000	0.0000000000000000	0.5000000000000000
0.6952342785779840	0.4539374361406396	0.6946749577520503
0.3047657214220160	0.5460625638593604	0.3053250422479497
0.8047657214220160	0.5460625638593604	0.1946749577520503
0.1952342785779839	0.4539374361406396	0.8053250422479497
0.3047657214220160	0.9539374361406396	0.3053250422479497
0.6952342785779840	0.0460625638593604	0.6946749577520503
0.1952342785779839	0.0460625638593604	0.8053250422479497
0.8047657214220160	0.9539374361406396	0.1946749577520503
0.4936690208802648	0.7500000000000000	0.9074751120548148
0.5063309791197352	0.2500000000000000	0.0925248879451852
0.0063309791197352	0.2500000000000000	0.4074751120548148
0.9936690208802648	0.7500000000000000	0.5925248879451852

Comment 3:

The similarity in Raman spectra between the two structures, despite the significant difference in atomic masses between Sr and Pb, is surprising to me. While the authors have proposed theoretical explanations, it is advisable to investigate CsEuBr₃, which lacks an LPE and adopts the perovskite structure, to further validate their theory.

Response: The similarity in the Raman spectra is because in the low-frequency region where the Raman signals are most prominent, no significant changes of the vibrations occurred. Other regions are changed more in line with the reviewer's assessment. We have revised the text to better highlight this aspect. While we appreciate the suggestion and interest in our work, investigating an entirely new material (CsEuBr₃) goes beyond the scope of our present manuscript. We believe this idea by the reviewer is very interesting and view it as a topic for future work. We have changed the manuscript to include this as an outlook.

Changes:

On p. 4 we now write:

The similar phonon DOS and the contributions of the M-site at low frequencies explain the limited shift of the CsPbBr₃ Raman spectrum, which might be surprising at first sight given the different atomic masses of Sr and Pb.

On p. 6, we now write:

Furthermore, investigating related compounds with distinct electronic configurations on the octahedral cation, such as CsEuBr₃, may provide further insight about chemical trends in tuning of the HaP properties.

Comment 4: In Figure 4, there appears to be a noticeable difference in Raman intensity between CsSrBr₃ and CsPbBr₃, as indicated by the signal-to-noise ratio. Could the authors clarify this observation and provide a comparison of Raman intensities between the two structures in their theoretical simulations?

Response: The reviewer is correct and we gladly explain this further: the scattering cross-section of CsSrBr₃ is notably weaker than that of CsPbBr₃, which is because a powder sample has been used in the higher-temperature measurements of CsSrBr₃ and because this material has a significantly higher bandgap. This higher bandgap leads to a weaker dielectric response at the Raman excitation wavelength (785 nm). We have revised the main text accordingly.

Changes:

On p. 5 of the revised version, we now write:

We note that differences in Raman intensity imply that the scattering cross-section of CsSrBr₃ is notably weaker than that of CsPbBr₃, which is due to its significantly higher bandgap and weaker dielectric response at the Raman excitation wavelength (785 nm) and because a powder sample of CsSrBr₃ has been used for which scattering of light in the back-scattering direction is considerably lower. The presence of a central peak ...

Comment 5:

The authors have suggested that the presence of an LPE in CsPbBr₃ enhances low-frequency octahedral tilting. However, it remains unclear why octahedral tilting and B-cation displacement

are correlated, given that these two types of distortions are typically mutually repulsive from a geometrical perspective.

Response: We agree with the reviewer that geometrically and chemically, B-cation displacements and octahedral tiltings should interact. However, in the halide perovskites these interactions are significantly different than in oxides, likely because of the sizable polarizability of halides such as I or Br. A consequence is that octahedral tiltings perturb the resonant-bonding network of X-Pb-X in these materials, as shown in previous work by the authors (Adv. Sci. 9, 2200706 (2022); ref. 17 in the paper). This is why these motions appear correlated: if B-cation displacements induce octahedral tiltings and vice versa, these types of motions will appear positively correlated, which is what we found in this work and is in line with further previous work by the authors (Mater. Adv. 2, 4610–4616 (2021), ref. 32 in the paper). In CsSrBr₃ this correlation is weaker because such a network is present to a much lower degree when the LPE is absent. We realize this could have been explained in a better way and have revised the manuscript accordingly.

Changes:

We now write, on p. 6 of the revised version:

M-site displacements and octahedral rotations are correlated because the latter is accompanied by changes of the Br-Pb-Br resonant network¹⁷ affecting the charge density in the vicinity of the M-site. While this effect is very weak in CsSrBr₃ (see Supplementary Information), the non-zero $C_{\alpha\beta}$ for this case shows that the presence of a LPE is not necessary to couple octahedral rotations and M-site displacements because the ions are still interacting through other types of interactions, e.g., electrostatically or due to Pauli repulsion. In CsPbF₃, the interaction of tilting and M-site displacements is strong enough to drive the adoption of an unusual tilt pattern.³⁶

Comment 6:

Previous studies have demonstrated an anharmonic and even a double-well potential energy surface for B-cation displacement in cubic CsPbBr₃ upon thermal heating, J. Am. Chem. Soc. 2016, 138, 11820. This is expected to contribute to lattice anharmonicity. Could the authors clarify whether and how this factor may contribute to the differences between the two structures?

Response: To address the question, we used the MD calculations to extract dynamic potential wells of M-site displacements for the two compounds (525 K for CsPbBr₃ and 570 K for CsSrBr₃) via a Boltzmann-inversion technique. The figure below shows the result for Pb and Sr: first, both datasets can be fit almost perfectly with a function that is quadratic in the displacements, showing that the potential associated with M-site displacements is harmonic. Second, while both potentials are very similar the Pb displacements appear to follow a somewhat softer potential owing to its larger mass. Taken together, we think these aspects are not very relevant for the discussion and findings in our work and prefer to include them here for reviewing purposes only.

Figure R1: **Dynamic potential wells of CsPbBr₃ and CsSrBr₃.** Potential energy as a function of the M-site atomic displacement along the x direction. The potential is obtained from an inversion technique that uses the MD-calculated histograms of atomic displacements and Boltzmann statistics. We find that for M=Pb and M=Sr the change in potential energy (blue and orange datapoints) can be fit well by a quadratic function (blue and orange line), showing that the potential-energy change is harmonic. The result is similar for both materials.

Comment 7: I just found that quartz crystal exhibits low-frequency Raman intensity at high temperatures, see J. Chem. Phys. 56, 1528–1533, 1972. A control experiment is recommended. Additionally, in Figure S3, the baselines on the anti-Stokes side of the high-temperature spectra appear higher than those on the Stokes side. What is the reason for this discrepancy?

Response: This is correct, but the published quartz contribution towards zero frequency in the literature is narrower in frequency than our observation. We performed a control experiment showing that there is no significant scattering from the capillary. The most pronounced peak at 110 cm^{-1} of quartz does not appear in our Raman spectrum. We have added this important information to the methods section of the revised manuscript and the figure for the control experiment to the Supplemental Material. Regarding the anti-Stokes to Stokes ratio at elevated temperatures, we note that the system response becomes noticeable at very high temperatures. This is due to a substantial increase in the population of phonons, leading to a rise in the anti-Stokes signal. Importantly, however, we do not derive any findings from the Stokes-to-anti-Stokes ratio, and thus the impact of this aspect is not critical for our analysis.

Changes:

On p. 9, we now write:

We note that while Raman spectra on quartz show a contribution towards zero frequency,⁸¹ it is narrower in frequency than what we observe. Results from control experiments (see Supplemental Material) show that the main signals from quartz do not contribute to the measured Raman spectra of CsSrBr₃.

On p. 5 of the Supplemental Material we included the following figure:

Fig. S5: **Control experiment for quartz.** Raman spectrum of quartz capillary compared to the Raman spectra of CsSrBr₃ at various temperatures. The main feature in the Raman spectrum of quartz at $\sim 110\text{ cm}^{-1}$ is not visible in the spectra of CsSrBr₃.

Comment 8:

Central peaks have been observed in other materials, particularly oxide perovskites. It would be valuable to compare the findings in halide perovskites with those observed in other systems to gain a broader perspective on the phenomenon.

Response: We agree with the reviewer and notice recent work by some of the authors in which such a comparison has been conducted. We agree that it is insightful to refer to this analysis and have revised the manuscript accordingly.

Changes:

On p. 6 of the revised manuscript, we now write:

*We note that recent work by some of the present authors has investigated the commonalities and differences between oxide perovskites and HaPs in this context.*⁴⁴

Reviewer 2:

Comment 1:

The manuscript “Disentangling the Effects of Structure and Lone-Pair Electrons in the Lattice Dynamics of Halide Perovskites” is an interesting look at the role of the metal’s electron configuration in the lattice dynamics of a halide perovskite. The experimental work is well done, as is the theory. My main critique is the framing of the article—the similar behavior between metals with noble gas and d10s2 electron configuration is not surprising to this reviewer, and indeed is what should be expected from the theory surrounding Jahn-Teller and pseudo-Jahn-Teller distortions.

Response: We thank the reviewer for noting that our research work was well-done and gladly address the remaining comments below.

Comment 2:

The (SrBr₆)⁴⁻ cluster has the same electron configuration as the (TiO₆)⁸⁻ cluster, which is present in many oxide perovskites such as SrTiO₃ and BaTiO₃. Ti⁴⁺ has a noble gas electron configuration, just like Sr²⁺, so the behavior of the metals should be very similar. (PbBr₆)⁴⁻ undergoes similar pseudo-Jahn-Teller distortions, which the authors of this manuscript know well (e.g., (1) Fabini, D. H.; Seshadri, R.; Kanatzidis, M. G. MRS Bulletin 2020, 45 (6), 467–477. <https://doi.org/10.1557/mrs.2020.142>). I suggest Bersuker, I. B. Chemical Reviews 2013, 113 (3), 1351–1390. <https://doi.org/10.1021/cr300279n> as a good reference, specifically Table 6.

Response: We appreciate the reviewer’s engagement with this point and suggestion to connect our framing more directly to the terminology and literature on pseudo-Jahn–Teller (PJT) effects. We agree that main-group cations with ns^2 electron configurations are subject to either a “weak” or a “strong” PJT, depending on the energy of the first-excited state in the reference configuration and the strength of the (linear) vibronic coupling, as mediated by covalency with the anions. We have revised our language and framing at several points in the manuscript (highlighted below) to clarify the connection between these ideas, and cited the excellent review of Bersuker.

However, we disagree with the assertion that Sr²⁺ and Ti⁴⁺ should be expected to behave similarly merely since they share a noble gas electron configuration. In almost every other respect, these two ions differ greatly, and this is reflected in the very different coordination environments they tend to adopt across the corpus of known inorganic compounds. Notably, the lowest excited states for octahedral [TiO₆]⁸⁻ are low-lying degenerate levels derived from the Ti 3*d*–O 2*p* interaction, hence the strong PJT effect and the resulting acentric environments observed in many titanates(IV), with the particulars depending on ligand hardness and chemical pressure. On the other hand, oxides of alkaline earth dications like Sr²⁺ exhibit extremely wide bandgaps [Phys. Rev. 111 (1958) 113–119] due to their shallow *ns* orbitals – there is no pseudodegeneracy to drive the PJT since the lowest-lying excited states are so high relative to the ground state, regardless of their symmetries. The case of strong vibronic coupling, which in principle could nonetheless allow for a pronounced PJT effect despite the large energy separation to degenerate excited states of appropriate symmetry (as in NH₃), is ruled out for alkaline earth bromide perovskites: Our first-principles calculations for CsSrBr₃ reveal negligible covalency (Figure 1f) and others reported for CsCaBr₃ by one of the present authors reveals no PJT instability even under extreme tensile strain [J. Am. Chem. Soc. 138 (2016) 11820–11832].

Changes:

The framing and discussion of bonding and structural instabilities has been modified throughout the manuscript. Rather than listing all changes in an excessive list, we provide snapshots to illustrate the modified framing. For example, portions of the introduction now read:

[...] This particular aspect of HaPs leads to a “strong” or “weak” pseudo-Jahn–Teller (PJT) effect,^{27–29} depending on the particulars of cation and anion composition and chemical pressure. The “weak” case influences local structure,^{30,31} lattice dynamics³² and ionic dielectric responses,^{26,30,33,34} while the “strong” case additionally results in the formation of a stereochemically-expressed electron lone-pair and impacts average crystal structures.^{32,35–37} [...]

[...] Both exhibit almost identical geometrical and structural parameters, but CsSrBr₃ exhibits a negligible PJT effect on the octahedral Sr²⁺ site, owing to weaker vibronic coupling to degenerate excited states of appropriate symmetry which lie higher in energy than in the Pb²⁺ case, allowing separation of the effects of the ns^2 electron configuration and the geometry on the lattice dynamics in a direct manner. [...]

and a portion of the discussion now reads:

[...] Notably, CsGeBr_3 , which exhibits no octahedral tilting transitions⁶⁸ and a broad Raman central peak in the cubic phase with a mode reflecting persistent pyramidal $[\text{GeBr}_3]^-$ anions,³² corresponds to the “strong” PJT²⁹ case: Stereochemically expressed cation lone-pairs are evident in the low temperature average structure⁶⁸ and in the local fluctuations of the cubic phase.³² Dynamic symmetry-breaking giving rise to a broad Raman central peak is thus observed for three distinct bonding regimes with regard to pseudo-Jahn–Teller effects: strong PJT (CsGeBr_3),³² weak PJT (CsPbBr_3 and others),³³ and negligible PJT (CsSrBr_3). [...]

Comment 3: The framing of the anharmonic effects in APbX3 halide perovskites as a function of the s lone pair is likely the cause of this confusion. The term “stereochemically active lone pair” is common in the literature, but in my mind is misleading because the lone pair itself is not the cause of the distortions—they originate from degeneracies in the first excited state/the pseudo-Jahn-Teller effect, which is why similar distortions occur in CsSrBr_3 , just like they do in ATiO_3 oxide perovskites. This confusing nomenclature is discussed in a few places in the recent halide perovskite literature—for instance, in Straus, D. B.; Mitchell Warden, H. E.; Cava, R. J. *Inorganic Chemistry* 2021, 60 (17), 12676–12680. <https://doi.org/10.1021/acs.inorgchem.1c01277>.

Response: We thank the reviewer for the helpful suggestions on improving clarity and terminology. As above, we agree on the connection between the PJT effect (general) and stereochemical expression of main-group cation lone pairs (manifestation of a strong PJT for ions in a particular subset of electronic configurations). We have adapted our framing to emphasize that this study directly examines influence of a “weak” PJT effect (ubiquitous among the most technologically-relevant Pb perovskites) on structural dynamics by comparison to a system with negligible PJT effects. We further place this in the context of existing work from some of the present authors on the case of a “strong” PJT effect [*Mater. Adv.* 2 (2021) 4610-4616].

However, as above, we still disagree that “similar distortions occur in CsSrBr_3 , just like they do in ATiO_3 oxide perovskites.” CsSrBr_3 exhibits none of the hallmarks of a strong PJT (absence of an inversion center at the Sr^{2+} site which does not have an “innocent” origin in packing considerations, or loss of one on cooling) nor of a weak PJT (softening of infrared-active modes that would break inversion at the cation site for a stronger PJT, leading to elevated ionic dielectric response) [Bersuker, *The Jahn-Teller Effect*, 2006, Chapter 4: Pseudo Jahn–Teller, product Jahn–Teller, and Renner–Teller effects]. CsPbBr_3 , on the other hand, is an excellent example of the weak PJT, and it is this distinction between the two compounds which allows us to directly test the influence of the weak PJT on the structural dynamics and optoelectronic properties of the technologically-relevant Pb(II) bromide and iodide perovskites.

We have revised the manuscript to reflect how our study addresses a key gap in the knowledge of structural dynamics and optoelectronic properties of halide perovskites as a function of chemical bonding: The previously reported case of a strong PJT effect (CsGeBr_3 , with a stereochemically-expressed lone-pair) [*Mater. Adv.* 2 (2021) 4610-4616] and our report of a weak PJT effect (CsPbBr_3 , with $6s^2$ cations but without lone-pair formation in the average structure) and a negligible PJT effect (CsSrBr_3).

Changes:

The framing and discussion of bonding and structural instabilities has been modified throughout the manuscript. See the prior response for example passages.

Comment 4:

This manuscript should be published in Nature Communications after being revised to highlight the similar role of the pseudo-Jahn-Teller effect in both APbX₃ and ASrX₃ halide perovskites as well as emphasizing the similar electron configurations of the titanite oxide perovskites with CsSrBr₃.

Response: We think that the additional clarifications in the revised version have improved the discussion of our findings. We thank the reviewer for their remaining feedback and the strong support for our work (“This manuscript should be published in Nature Communications”).

Reviewer 3:

Comment 1: This paper investigates the relationship between lattice anharmonicity and presence of lone-pair electrons (LPE) in metal halide perovskites (HaPs) by combining first-principal simulations with IR and Raman spectroscopy measurements on CsPbBr₃ and CsSrBr₃.

The central result that the authors establish is that “the presence of a lone-pair electrons (LPE) is not required for the strong anharmonicity in the low-energy lattice dynamics of soft HaP semiconductors.” This result is established mainly based on computational modeling, partially validated with Raman spectroscopy. This result confirms prior conclusions previously reported in the literature (namely the fact that LPE is not required for strong anharmonicity in HaPs), which is appropriately cited by the authors. While the authors here confirm this conclusion by means of a comparison between CsPbBr₃ (LPE) and CsSrBr₃ (no LPE), I do not find that this provides sufficiently new insights to warrant the high visibility of a publication in Nature Communications. The two compositions studied show great similarity in crystal structure and phase transition sequences, and Sr lacks active lone-pair electrons (LPE) compared to Pb. The observation of a similar central peak in Raman spectra for both compositions indicates that LPE are not necessary for anharmonic octahedral tilting. In general, the information from the paper is quite simple and straightforward, but the results are not surprising or really novel.

Response: We thank the reviewer for their critical feedback, which prompted several changes to the manuscript and additional research work. First, while the effect of LPE on anharmonicity and central peaks has been discussed before, this manuscript is the first to provide unambiguous findings through complete removal of the LPE when synthesizing and investigating single crystals of CsSrBr₃. We have also performed extensive theoretical work to precisely understand what the origins of anharmonicity are, providing further crucial insights. Because chemical tuning of anharmonicity in these materials as expressed through the central peak is a topical issue for halide perovskites and beyond, we believe these findings are novel and important. Second, some of the below comments have addressed the relation between theoretical and experimental data and raised a possible matter of concern. We have clarified these aspects and performed detailed additional theoretical calculations to compute the Raman spectra of both compounds directly from MD. Our new results strongly support our main findings and interpretations. Together with extensive text revisions that streamline the discussion and clarify open questions by all reviewers, we think these changes improved our work and hope that the reviewer will find our revised manuscript worthy of publication in Nature Communications.

Comment 2: Overall, the main point that LPE is not necessary for the presence of low-frequency anharmonic octahedral rotations is solid, based on simulations and the observation of a central Raman peak in cubic CsSrBr₃. However, I disagree that the simulations in the orthorhombic phase show "good agreement" with the measurements. Perhaps the MD simula-

tions need to be better benchmarked by comparing with previously reported experimental data or other simulations.

Response: The reviewer has raised a possible point of concern when theoretical calculations and experiments were compared in our work. The reviewer asked whether the comparison of the experimental Raman spectra (at 80 K) and the DFT calculations (based on harmonic phonons) requires further benchmarks because there is poor agreement.

In response, we first discuss the original findings in more detail: Fig. 3b shows that CsPbBr₃ exhibits three broader features at frequencies below and one feature with a weaker intensity at frequencies above 100 cm⁻¹ in the experimental measurements. The data for CsSrBr₃ exhibit a structured feature around 50 cm⁻¹, a feature close to but below 100 cm⁻¹, and then three weaker signals for frequencies between 100–150 cm⁻¹. This is in agreement with what can be deduced from the original theoretical spectra shown in Fig. 3d: they find three features at frequencies below 100 cm⁻¹ and a weak signal above it for CsPbBr₃, and a structured feature at 50 cm⁻¹, a signal close to 100 cm⁻¹, as well as three weaker signals between 100–150 cm⁻¹ in case of CsSrBr₃.

The one feature for which there appears to be a significantly larger shift in the comparison, is the weak signal above 100 cm⁻¹ in CsPbBr₃, which appears at ~140 cm⁻¹ in experiment and at ~120 cm⁻¹ in the calculations. While this shift in the comparison is slightly larger than those of the other features, it is still reasonable considering that perfect agreement is not expected between experiment and theory especially for relative Raman intensities. Overall, we consider this to be in good agreement and the original statement fair. We agree with the reviewer that a more detailed comparison is helpful and updated the manuscript accordingly.

Secondly, this question by the reviewer and their below comments have prompted extensive additional theoretical work. We have performed extensive additional theoretical calculations for computing the finite-temperature Raman spectrum from MD. As a result, we now show new data that report broad central peaks in both compounds also from these theoretical calculations. Akin to the experimental findings contained already in the original manuscript, these new results support our analysis and interpretation and make our manuscript stronger overall.

Changes:

On p. 4 of the revised manuscript we now write:

Specifically, the experimental spectrum of CsPbBr₃ in Figure 3b finds three broader features at frequencies below and one weaker-intensity feature at frequencies above 100 cm⁻¹. Conversely, CsSrBr₃ exhibits a structured feature around 50 cm⁻¹, a pronounced signal close to 100 cm⁻¹, and then a series of weaker intensities between 100–150 cm⁻¹. While the DFT-computed Raman activities calculated in the harmonic approximation are in broad agreement with these findings (see Figure 3d), we note a slightly larger deviation of approximately 20 cm⁻¹ for the higher-frequency peak in CsPbBr₃. These findings lead us to conclude that unlike in IR ...

We updated Fig. 4 with the below figure on p. 5 of the revised manuscript. It now contains new data showing theoretical Raman spectra from MD. We decided to now show a different temperature for the experimental Raman spectrum of the Pb-compound to facilitate the experiment-theory comparison.

Fig. 4: **Lattice dynamics at higher temperature.** Raman spectra of CsPbBr₃ (panel a) and CsSrBr₃ (panel b) in the high-temperature cubic phase measured experimentally and calculated using DFT-MD. The central peak appears for both compounds in the experiments and computations despite significant differences in bonding: [PbBr₆]⁴⁻ is proximate to lone-pair formation (*i.e.*, exhibits a “weak” PJT effect),²⁹ while PJT effects associated with [SrBr₆]⁴⁻ are negligible

We have added a new section to the Methods section on p. 8 of the revised main text describing the Raman calculations from MD:

Raman Spectra From Molecular Dynamics

*DFT-based MD calculations were used to compute the high-temperature Raman spectra of CsPbBr₃ and CsSrBr₃. We calculated Raman intensities from the autocorrelation function of the polarisability, as detailed elsewhere.⁸⁵ The polarizabilities were calculated with DFPT⁸³ on 400 evenly-spaced snapshots every 0.11 ps for a total of 44.8 ps. The *k*-grid employed for the DFPT calculations was set to 4 × 4 × 4 after testing convergence of the polarisability tensor for several snapshots.*

Comment 3: The paper could also benefit from a deeper investigation of the simulations, to clarify how LPE could interact with anharmonic low-frequency modes, and thus result in stronger Cab(w) in CsPbBr₃ than CsSrBr₃.

Response: The correlation is because the *ns*² electron configuration of Pb²⁺ implies formation of a resonant-bonding network involving Br-Pb-Br ions in HaPs, discussed in previous work by some of the authors (Adv. Sci. 9, 2200706 (2022); ref. 17 in the paper). As shown in this study, octahedral rotations perturb the resonant-bonding network and, hence, the charge density in

the vicinity of the M-site. Hence the octahedral and M-site displacement motions are more correlated in CsPbBr₃ than in CsSrBr₃. This can be better explained and we changed the main text accordingly (see also response to comment 5 by reviewer 1). We also added new theoretical data on finite-temperature Raman spectra (see response to comment 2 by this reviewer).

Changes:

We extended the discussion on correlated octahedral and M-site motions on p. 5 of the revised manuscript, which now reads:

M-site displacements and octahedral rotations are correlated because the latter is accompanied by changes of the Br-Pb-Br resonant network¹⁷ affecting the charge density in the vicinity of the M-site. While this effect is very weak in CsSrBr₃ (see Supplementary Information), the non-zero $C_{\alpha\beta}$ for this case shows that the presence of a LPE is not necessary to couple octahedral rotations and M-site displacements because the ions are still interacting through other types of interactions, e.g., electrostatically or due to Pauli repulsion. In CsPbF₃, the interaction of tilting and M-site displacements is strong enough to drive the adoption of an unusual tilt pattern.³⁶

Comment 4: The discussion feels diluted, reiterating points already made in intro and results.

Response: Our goal for the discussion was to place the work into context of previous findings, which required us to reiterate the main findings and motivation. We agree with the reviewer that there were too many redundancies and that the discussion should be streamlined. We have considerably revised and shortened it.

Changes:

We streamlined the discussion, on p. 6 of the main text, where the changed part now reads:

We directly disentangled structural and chemical effects in HaPs by comparing CsPbBr₃ and CsSrBr₃, two compounds with similar ionic radii and structural properties but entirely different orbital interactions that leave CsSrBr₃ without the ability to form a LPE. While formation of a LPE is paramount for the optoelectronic properties of these materials, using the Raman central peak at higher temperatures as a fingerprint to detect anharmonicity we found it to appear also for the LPE-absent CsSrBr₃ and to correlate with slow, anharmonic rotations of the octahedra.

Comment 5: Overall, I find that the results in the manuscript are a bit too thin, and the impact too limited, to warrant publishing in Nature Communications. I recommend shortening the paper and publishing in a more specialized journal.

Response: The constructive criticism by the reviewer helped us to improve the quality of our work. The agreement between experiment and theory already contained in the original manuscript has now become more apparent through improved explanations. We hope that our additional discussion has been streamlined to better highlight the novelty of the work. Finally, the addition of new theoretical data demonstrating Raman central peaks in both compounds from fully-anharmonic MD supports our conclusions and makes the manuscript stronger. Altogether, our study provides an unambiguous and novel analysis of one of the key chemical effects in a topical material class, which has been achieved through a combined synthesis-experiment-computation effort. For these reasons, we believe it is well-suited for broad dissemination and hope that the reviewer will find the revised version suitable for Nature Communications.

Further changes:

- Manuel Grumet has been added as an author because he assisted in the new research work on MD-Raman calculations which have been included in the revised version.
- Several references were added to augment the discussion on pseudo-Jahn–Teller effects.

REVIEWER COMMENTS

Reviewer #1 (Remarks to the Author):

The revision has addressed my previous comments, and I support publication of this exciting manuscript.

Reviewer #2 (Remarks to the Author):

I disagree with some of the authors' responses to my review comments and cannot support publication until they are appropriately addressed. I still believe that the paper is well done and should be published in Nature Communications once these issues are addressed to both the authors' and my satisfaction. Resolving this disagreement will result in a stronger manuscript, whether I or the authors are correct.

It is not clear to me how the difference in band gap between CsPbBr₃ (~2.3 eV) and CsSrBr₃ (~4+ eV) makes any difference in the applicability of the pseudo-Jahn-Teller effect and how this difference causes the authors to conclude that CsSrBr₃ does not exhibit the pseudo-Jahn-Teller effect. In both cases, there is effectively no thermal excitation of carriers from the valence to the conduction band at room or elevated temperatures since kT (0.026 eV at 300 K) is much much smaller than the band gap.

Furthermore, the oxide perovskite SrTiO₃ is a white solid like CsSrBr₃ with a band gap of ~3.5 eV (Lee, C.-H. et al. Effect of reduced dimensionality on the optical band gap of SrTiO₃. Applied Physics Letters 102, 122901 (2013)), between that of CsPbBr₃ and CsSrBr₃, so the authors' argument that oxide and halide perovskites behave differently because of low-lying degenerate excited states is not correct. The applicability of the authors' argument that Ti⁴⁺ and Sr²⁺ are expected to behave differently in perovskite compounds. They have similar band gaps and identical electronic configurations. In addition, SrZrO₃ is also an orthorhombic GdFeO₃-type perovskite at room temperature with a band gap exceeding 5 eV (Cavalcante, L. S. et al. SrZrO₃ powders obtained by chemical method: Synthesis, characterization and optical absorption behaviour. Solid State Sciences 9, 1020–1027 (2007)), and Zr⁴⁺ has exactly the same electronic configuration as Sr²⁺ does.

It is certainly possible that I am incorrect—I am happy to be proven wrong about this. If the authors still believe that I am incorrect, I need to see an appropriately referenced response to this comment (including references that support their arguments that CsSrBr₃ will behave differently than CsPbBr₃, SrTiO₃, and SrZrO₃) so I can properly evaluate their arguments on this issue. This response letter contains no references pertaining to oxide perovskites.

I also take issue with the framing of the “strong” and “weak” forms of the pseudo-Jahn-Teller effect here, regardless of how it is used in other literature. My opinion is that the difference between the supposed “strong” and “weak” versions in these perovskites only depends on structural parameters, such as those incorporated into the Goldschmidt tolerance factor, rather than anything intrinsic about the compound itself. For instance, CsGeBr₃ has a Goldschmidt factor of 1.00, so it would be expected to be cubic, whereas CsPbBr₃ and CsSrBr₃ both have Goldschmidt factors of 0.86, and it thus would be expected to be lower symmetry and exhibit octahedral tilts. This has nothing to do with the pseudo-Jahn-Teller effect itself, only how it is represented because metal off-centering/stereochemically active lone pair is not required to occur by symmetry in an orthorhombic 3D perovskite because the

octahedral tilts break the relevant electronic degeneracies that are present in the cubic phase.

I am not qualified to comment the technical aspects of the theory concerns raised by Reviewer 3. In the context of the experimental findings, I think the theory makes sense and the alignment between the two results is as good as one can hope to find from DFT.

Reviewer #3 (Remarks to the Author):

The paper underwent a thorough review from three reviewers, who each brought up significant criticisms and requests for changes/additions. The authors have significantly improved their manuscript, and added valuable new results (Fig S2/Table S1) and new MD simulations of Raman spectra, in particular. The added discussion of comparison with pseudo-Jahn-Teller effect is also welcome. I now feel that the manuscript is suitable for publication in Nature Communications.

Reviewer 2:

Comment 1: I disagree with some of the authors' responses to my review comments and cannot support publication until they are appropriately addressed. I still believe that the paper is well done and should be published in Nature Communications once these issues are addressed to both the authors' and my satisfaction. Resolving this disagreement will result in a stronger manuscript, whether I or the authors are correct.

Response: We appreciate the reviewer's positive assessment that our work is "well done" and "should be published in Nature Communications once these issues are addressed [...]" We particularly appreciate the reviewer's commitment to resolving our points of disagreement and openness on this matter. We are happy to provide clarification of our claims, both here and in the manuscript, and make our argumentation around the pseudo-Jahn–Teller (PJT) effect more direct and quantitative.

As noted below, our most significant modification to the manuscript is the addition of a section in the Supplementary Information (referenced in the main text) which explicitly parameterizes the standard two-level PJT model using DFT total energy calculations on four series of compounds: Cs(Ge,Sn,Pb,Sr)Br₃, Ba(Ti,Zr,Sn)O₃, Sr(Ti,Zr,Sn)O₃, and (Ge,Sn,Pb,Sr)Te. The results of this additional effort are entirely consistent with our previous argumentation and claims, but allow us to more explicitly demonstrate the wide variation of pseudo-Jahn–Teller effects in these compounds. This in turn clarifies the rationale behind the particular chemical comparison we designed in order to assess the impacts of vibronic anharmonicity on the lattice dynamics of halide perovskites.

In developing this new section, we relied heavily on chapters 2, 3, 4 of the excellent book by Bersuker [Bersuker, *The Jahn-Teller Effect*, 2006, Cambridge University Press], especially Sections 2.1, 2.4, and 4.1. This book, a review [Bersuker, *Chem. Rev.* *113*, 1351-1390 (2013)], and two original research articles relevant to the PJT effect for s^2 cations [Van der Vorst & Maaskant, *J. Solid State Chem.* *34*, 301-313 (1980); Maaskant & Bersuker, *J. Phys. Condens. Matter* *3*, 37-47 (1991)] are already referenced in our previous revised manuscript. In this revision, we have additionally cited the original research work which uncovered the symmetry of the PJT effect for d^0 cations (*e.g.* ferroelectric oxides like BaTiO₃) [Bersuker, *Phys. Lett.* *20*, 589-590 (1966)] to strengthen the comparison to oxides.

Changes:

[in main text]

We disentangle structural and chemical effects in the lattice dynamics of HaP by comparing the well-known CsPbBr₃ with the far less studied CsSrBr₃. Both exhibit almost identical geometrical and structural parameters, but CsSrBr₃ exhibits a negligible PJT effect on the octahedral Sr²⁺ site, owing to **substantially** weaker vibronic coupling to degenerate excited states ~~of appropriate symmetry which lie higher in energy~~ than in the Pb²⁺ case (see Supplementary Information), allowing separation of the effects of the ns^2 electron configuration and the geometry on the lattice dynamics in a direct manner.

[new section in Supplementary Information]

Vibronic anharmonicity and the strength of the Pseudo-Jahn–Teller effect

The strength of the pseudo-Jahn–Teller (PJT) effect differs considerably between CsSrBr₃ and CsPbBr₃, as we demonstrate here. The energies of the two-level PJT problem with linear vibronic coupling (see Bersuker Section 4.1 for background and derivation) [S1] can be written as:

$$\epsilon_{\pm}(Q) = \pm\Delta + \frac{1}{2}(K_0 \pm F^2/\Delta)Q^2 \mp \frac{1}{4}(F^4/\Delta^3)Q^4 + \dots, \quad (1)$$

where Q is the nuclear distortion coordinate ($Q = 0$ in the high symmetry reference case, *i.e.* octahedral coordination), K_0 is the primary force constant in the absence of the vibronic interaction, F is the off-diagonal linear vibronic coupling constant, and 2Δ is the energy separation between the electronic ground and (degenerate) excited states in the reference case. The so-called “strong” case of the PJT effect, which produces a spontaneous distortion that breaks inversion symmetry, occurs when $\Delta < F^2/K_0$. Otherwise, in the so-called “weak” case, there is no spontaneous distortion, but the effective force constant of the *ungerade* nuclear displacements in the ground state is nonetheless vibronically softened by an amount F^2/Δ and *vibronic anharmonicity* is introduced (distinct from proper anharmonicity, see Section 2.4 of Bersuker). [S1] The magnitudes of these vibronic impacts on the ground state (softening of the quadratic force constant, anharmonicity) take on a continuum of values in various systems. We designed the comparison between CsPbBr₃ and CsSrBr₃ to test the influence of the vibronic anharmonicity on the lattice dynamics of halide perovskites – despite both formally exhibiting the weak PJT effect, as we show below, the vibronic contributions are markedly different in magnitude in these two systems, with CsPbBr₃ lying much closer to a spontaneous acentric distortion.

FIG. S1. **Energetics of trigonal distortions.** DFT-computed energies as a function of trigonal displacement of the octahedral cations, u_{111} , and fitted 2-level PJT model parameters for bromide perovskites (top row), oxide perovskites (middle rows), and rocksalt tellurides (bottom row). In the left panels, markers are data and solid lines are fits to Equation S2. Fitted values of a_2 and a_4 are visualized in the right panels across each chemical series. For each family of compounds except the Sr-oxides, a progression from negligible to weak to strong PJT effects is found as a_4 increases and a_2 decreases. For the Sr-oxides, the progression is similar but a_2 does not quite cross 0 for paraelectric SrTiO₃, unlike for ferroelectric BaTiO₃, illustrating the secondary influence of chemical pressure. A strong PJT effect ($a_2 < 0$) is found for systems with acentric cation site symmetry ($3m$) in the experimental ground state crystal structures (CsGeBr₃, BaTiO₃, SnTe, GeTe, all space group $R\bar{3}m$). CsSnBr₃, recently shown to exhibit acentric Sn site symmetry (1 , space group $P2_1$) at low temperatures [S2] lies near $a_2 = 0$, with a weak PJT erroneously predicted at the equilibrium unit cell size. All other cases with $a_2 > 0$ have cations on sites with inversion symmetry in the experimental ground state crystal structures ($m\bar{3}m$ for BaSnO₃, BaZrO₃, SrTe, and PbTe; $\bar{1}$ for CsSrBr₃, CsPbBr₃, SrSnO₃, and SrZrO₃; $4/m$ for SrTiO₃). CsSrBr₃, (Ba,Sr)SnO₃, and SrTe exhibit negligible PJT effects: the vibronic anharmonicity coefficient, a_4 , is much smaller than for the other compounds in each series.

To parameterize this model in terms of observables from Born-Oppenheimer DFT, we rewrite the energy of the lower surface (electronic ground state) up to 4th order and drop the “–” subscript, yielding:

$$\epsilon(Q) + \Delta = a_2 Q^2 + a_4 Q^4, \quad (2)$$

where $a_2 = \frac{1}{2} (K_0 - F^2/\Delta)$ and $a_4 = \frac{1}{4} (F^4/\Delta^3)$. We fit a_2 and a_4 to DFT total energies from a series of distorted structures (using VASP, PAW potentials, PBE functional, 600 eV plane wave cutoff, $7 \times 7 \times 7$ k -point mesh for oxide and bromide perovskites, $9 \times 9 \times 9$ for rocksalt tellurides). It is not straightforward to extract the appropriate Δ from DFT so that F and K_0 can be obtained explicitly from a_2 and a_4 . Nevertheless, there is insight to be had in the values of a_2 and a_4 directly: $a_2 < 0$ corresponds to the strong case of the PJT effect, and $a_4 \rightarrow 0$ as the PJT effect becomes negligible (*i.e.*, either as the vibronic coupling constant $F \rightarrow 0$ or as the energy separation between the ground state and the degenerate excited states $2\Delta \rightarrow \infty$). Despite the differing symmetries of the degenerate excited state orbitals for d^0 and s^2 cations in octahedral coordination, these two cases are known to give rise to the same, trigonal distortion. [S3, S4] For this study of chemical trends and in a manner similar to previous reports, [S5, S6] we apply rigid trigonal displacements of the octahedral cations along [111] rather than perturbing along the exact phonon eigenvector.

We examine three series of compounds (bromide perovskites, oxide perovskites with either Ba^{2+} or Sr^{2+} on the A -site, and rocksalt tellurides) which modulate the PJT effect strength across a wide range, and correlate the results (Figures S1 and S2) to chemistry and bonding. Figure S1 shows that, as we traverse the series $\text{CsSrBr}_3 \rightarrow \text{CsPbBr}_3 \rightarrow \text{CsSnBr}_3 \rightarrow \text{CsGeBr}_3$, the quadratic coefficient in the 2-level PJT model, a_2 , softens considerably and eventually becomes negative, corresponding to the strong case of the PJT effect and the formation of lone pairs in CsGeBr_3 . Though atomic masses play a role via the primary force constants, K_0 , one can see this is not the dominant factor in the trend in a_2 : the vibronic contribution to the quadratic coefficient is essential. Simultaneously, the quartic coefficient, a_4 , (vibronic anharmonicity) rises considerably across the series. The value of a_4 for CsPbBr_3 is 8 times that for CsSrBr_3 . Thus, these two compounds are in very different PJT regimes and it is the impact of this distinction on the lattice dynamics of the two which we also probe in this study. We refer to CsSrBr_3 as exhibiting a “negligible” PJT effect, and contrast its behavior with that of CsPbBr_3 , which shares a weak PJT effect with other technologically-relevant Pb(II) bromides and iodides, and with that of CsGeBr_3 , [S7] which exhibits a strong PJT effect.

FIG. S2. **Modulating the PJT effect.** Values of a_2 and a_4 from fitting DFT energies to Equation S2 across several bromide perovskites, oxide perovskites, and rocksalt tellurides. To illustrate the influences of the primary force constant, K_0 , and vibronic coupling constant, F , contours of K_0 at a fixed Δ are shown. Each family of materials falls roughly onto a curve of varying F , with the families differentiated from one another by different elastic properties (K_0). Notably, the series of bromide perovskites runs from a strong PJT effect (CsGeBr₃) through a weak PJT effect (CsPbBr₃) to a negligible PJT effect (CsSrBr₃) as the vibronic coupling coefficient, F , decreases towards zero. Additionally, the trends in a_2 show that softening of the harmonic force constant is not merely a mass effect, but includes a substantial vibronic contribution to the curvature. [S1]

Figure S1 reveals similar trends for the oxide perovskites and the rocksalt tellurides, with perfect correspondence between the finding of a strong PJT effect ($a_2 < 0$) and those systems which exhibit acentric cation environments in their ground state phases. Notably, the strengths of the PJT effects we fit numerically also perfectly track the chemical trends expected for d^0 cations (*e.g.* Ti⁴⁺ exists in more strongly distorted environments than Zr⁴⁺) [S8] and s^2 cations (*e.g.* distortions increase in the order Pb²⁺ < Sn²⁺ < Ge²⁺). [S9]

Figure S2 places the fitted PJT model parameters onto a unified scale to show how they vary between the chemical families. Fixed- Δ , varying- F curves of $(a_2(K_0, F, \Delta), a_4(F, \Delta))$ are shown for evenly spaced values of K_0 to illustrate that the families are primarily separated by their varying elastic properties in the absence of vibronic coupling (K_0), while intra-family variation can be roughly explained by differing vibronic coupling coefficients, F . Fixed- F , varying- Δ curves have a different functional form and fit the intra-family variation less well, suggesting the energy separation between the ground and degenerate excited states plays a secondary role in modulating the PJT effect strength across these particular series.

Comment 2:

It is not clear to me how the difference in band gap between CsPbBr₃ (2.3 eV) and CsSrBr₃ (4+ eV) makes any difference in the applicability of the pseudo-Jahn-Teller effect and how this difference causes the authors to conclude that CsSrBr₃ does not exhibit the pseudo-Jahn-Teller effect. In both cases, there is effectively no thermal excitation of carriers from the valence to the conduction band at room or elevated temperatures since kT (0.026 eV at 300 K) is much much smaller than the band gap.

Response: We regret that our previous explanation was incomplete with respect to energy separations and bandgaps. In short, there are 3 parameters which determine the strength of the PJT effect: 1) the energy separation between the electronic ground state and the excited states which are degenerate in the high symmetry (reference) case; 2) the linear off-diagonal

vibronic coupling coefficient which describes the degree of covalent mixing of these states under *ungerade* nuclear displacements which break inversion symmetry; and 3) the primary force constant (stiffness) of the *ungerade* mode in the fictitious absence of the vibronic coupling. Indeed, all else being equal, a greater energy separation to the relevant degenerate excited states (which is often implied by a larger bandgap but is not the same) weakens the PJT effect. However, on further investigation, variation of this energy separation may explain less of the difference in PJT effect strengths in this particular set of compounds than variations in the vibronic coupling coefficient. We have removed language about the energy separation at relevant points as detailed below, and emphasized the vibronic coupling coefficient.

To the reviewers' point about thermal excitation of carriers, we respectfully disagree. The PJT is an athermal phenomenon and does not rely at all on thermal excitation of carriers across a bandgap. Rather, under *ungerade* nuclear displacements from high symmetry, the vibronic coupling of nuclei and electrons mixes states which *would be* exclusively "ground" or "excited" in the high symmetry reference case – the relevance of the "excited" states is not because of thermal- or photo-excitation of carriers into these states.

In the newly added section of the Supplementary Information, we parameterize the two-level PJT model using DFT calculations, showing explicitly that CsGeBr₃ exhibits a "strong" PJT effect (with a spontaneous polar distortion of the Ge²⁺ environment and formation of a lone pair), CsPbBr₃ a "weak" PJT effect (no spontaneous distortion, but vibronic softening of the effective harmonic force constant and introduction of vibronic anharmonicity), and CsSrBr₃ a negligible PJT effect ($\sim 8\times$ smaller vibronic anharmonicity than CsPbBr₃). By parameterizing the 2-level PJT model with DFT, we find that the energy separations to the degenerate excited states are likely secondary to differences in the vibronic coupling coefficients in these compounds. To improve clarity, we have removed the reference to the energies of the excited states in the introduction, as detailed below. We regret this previous oversight.

Changes:

We disentangle structural and chemical effects in the lattice dynamics of HaP by comparing the well-known CsPbBr₃ with the far less studied CsSrBr₃. Both exhibit almost identical geometrical and structural parameters, but CsSrBr₃ exhibits a negligible PJT effect on the octahedral Sr²⁺ site, owing to **substantially weaker vibronic coupling to degenerate excited states of appropriate symmetry which lie higher in energy** than in the Pb²⁺ case (see Supplementary Information), allowing separation of the effects of the *ns*² electron configuration and the geometry on the lattice dynamics in a direct manner.

Comment 3:

Furthermore, the oxide perovskite SrTiO₃ is a white solid like CsSrBr₃ with a band gap of 3.5 eV (Lee, C.-H. et al. Effect of reduced dimensionality on the optical band gap of SrTiO₃. Applied Physics Letters 102, 122901 (2013)), between that of CsPbBr₃ and CsSrBr₃, so the authors' argument that oxide and halide perovskites behave differently because of low-lying degenerate excited states is not correct. the applicability of the authors' argument that Ti⁴⁺ and Sr²⁺ are expected to behave differently in perovskite compounds. They have similar band gaps and identical electronic configurations. In addition, SrZrO₃ is also an orthorhombic GdFeO₃-type perovskite at room temperature with a band gap exceeding 5 eV (Cavalcante, L. S. et al. SrZrO₃ powders obtained by chemical method: Synthesis, characterization and optical absorption behaviour. Solid State Sciences 9, 1020–1027 (2007)), and Zr⁴⁺ has exactly the same electronic configuration as Sr²⁺ does.

It is certainly possible that I am incorrect—I am happy to be proven wrong about this. If the authors still believe that I am incorrect, I need to see an appropriately referenced response to this comment (including references that support their arguments that CsSrBr₃ will behave

differently than CsPbBr₃, SrTiO₃, and SrZrO₃) so I can properly evaluate their arguments on this issue. This response letter contains no references pertaining to oxide perovskites.

Response: We appreciate the opportunity to clarify, and the reviewer’s openness on this point of disagreement. Our response to this comment mirrors our response above: 1) Bandgap is only a proxy for the minimum possible energy separation between the electronic states which are most strongly vibronically coupled; 2) Our newly added DFT parameterization of the 2-level PJT model reveals that energy separation may not be the dominant factor in the variation of PJT effect strengths between, *e.g.*, CsSrBr₃, CsPbBr₃, SrTiO₃, and SrZrO₃; 3) Therefore, we use our DFT calculations to more explicitly demonstrate that Sr²⁺ in CsSrBr₃ should *not* be expected to behave like Ti⁴⁺ or Zr⁴⁺ in SrMO₃. We regret our choice of bandgap as a proxy in our previous argumentation, though we reach the exact same conclusions through atomistic modeling as we did previously.

As shown in the fitted values of the lumped PJT model parameters, a_2 and a_4 , presented in Figure S2, Ti⁴⁺ in SrTiO₃ lies just to the weak side of the strong/weak PJT boundary and exhibits strong vibronic anharmonicity. This is precisely in line with its observed dielectric properties [Phys. Rev. 135, A748 (1964)], and with the observations that it can be coaxed into the ferroelectric state by small perturbations like achievable strains [Nature 430, 758-761 (2004)] or fields [Science 364, 1079-1082 (2019)].

From Figure S2, Zr⁴⁺ in SrZrO₃ exhibits a weaker PJT effect, with less pronounced vibronic anharmonicity and vibronic softening than for SrTiO₃. This is in accord with its more moderate dielectric response [J. Am. Ceram. Soc., 84 1648 (2001)] from less-elevated Born effective charges and less vibronically-softened IR mode frequencies. CsPbBr₃ also exhibits a weak PJT effect, with an a_4/a_2 ratio similar to that of SrZrO₃.

On the other hand, CsSrBr₃ exhibits a negligible PJT effect, with a_4 8x smaller than for CsPbBr₃. This is also in line with the substantial vibronic softening of the strongest IR mode of CsPbBr₃ relative to that of CsSrBr₃, which cannot be explained by the ion masses as demonstrated by the fictitious mass phonon calculations in the Supplementary Information. The influence on the lattice dynamics of the markedly different PJT effect strengths for [SrBr₆]⁴⁻ and [PbBr₆]⁴⁻, arising from the different bonding and orbital interactions in the two, is a key feature probed in our study.

Having shown that [SrBr₆]⁴⁻ exhibits a vastly weaker PJT effect than [PbBr₆]⁴⁻, [TiO₆]⁸⁻, or [ZrO₆]⁸⁻, we note that our PJT model calculations correctly predict the ground state structures of 13 of the 14 compounds considered: All four predicted to exhibit a strong PJT have acentric cation environments, while nine of the ten predicted to exhibit a weak PJT have centric cation environments. CsSnBr₃, recently shown by one of the authors to exhibit a polar ground state [arXiv: 2401.07978], is the sole incorrect prediction but lies very close to the weak/strong boundary in this simplified model.

Furthermore, we note that our PJT model calculations perfectly track the known chemical trends for both d^0 cations [Chem. Mater. 18, 3176-3183 (2006)], with distortion propensity increasing Zr⁴⁺ < Ti⁴⁺, and s^2 cations [Phys. Rev. B 67, 125111 (2003)], with distortion propensity increasing Pb²⁺ < Sn²⁺ < Ge²⁺. Examining our results for CsSrBr₃, SrTe, and (Ba,Sr)SnO₃, the model calculations also support our expectation that cations like s^0 Sr²⁺ and d^{10} Sn⁴⁺ exhibit negligible PJT effects (we are not aware of an appropriate citation here, but expect this is an unsurprising result to many chemists).

Indeed, it is the steadfastness of these chemical trends, which are implicit among the known crystal structures of inorganic materials, which led us to omit explicit modeling of the PJT effect in earlier versions of the manuscript. We hope that inclusion of the new section in the Supplementary Information addresses the concerns of the reviewer and improves the clarity of

our work.

Changes: [Same as above responses]

Comment 4:

I also take issue with the framing of the “strong” and “weak” forms of the pseudo-Jahn-Teller effect here, regardless of how it is used in other literature. My opinion is that the difference between the supposed “strong” and “weak” versions in these perovskites only depends on structural parameters, such as those incorporated into the Goldschmidt tolerance factor, rather than anything intrinsic about the compound itself. For instance, CsGeBr₃ has a Goldschmidt factor of 1.00, so it would be expected to be cubic, whereas CsPbBr₃ and CsSrBr₃ both have Goldschmidt factors of 0.86, and it thus would be expected to be lower symmetry and exhibit octahedral tilts. This has nothing to do with the pseudo-Jahn-Teller effect itself, only how it is represented because metal off-centering/stereochemically active lone pair is not required to occur by symmetry in an orthorhombic 3D perovskite because the octahedral tilts break the relevant electronic degeneracies that are present in the cubic phase.

Response:

We respectfully disagree with the reviewer here, and are happy to explain our reasoning. The PJT effect is a covalent phenomenon, deriving only from the interaction of a particular ion with its first coordination shell (long-range electrostatics, the spin-orbit interaction, and band dispersion are treated as perturbations atop this dominant feature). On the other hand, the Goldschmidt tolerance factor expresses only the relative sizes of fictitious but conceptually useful hard sphere ions as a means of predicting the existence and magnitude of octahedral tilts in perovskites.

Broadly speaking, the Goldschmidt tolerance factor is quite predictive, hence its ongoing and frequent use. But it pertains to a qualitatively different sort of distortion – octahedral tilting does break certain degeneracies, but it does not break those responsible for the PJT effect. In perovskites with tilting only, the octahedral cations remain on inversion centers (site symmetry $m\bar{3}m$ for cubic $Pm\bar{3}m$, $4/m$ for tetragonal $P4/mbm$ and $I4/mcm$, $\bar{1}$ for orthorhombic $Pnma$), and thus do not eliminate the particular vibronic interactions which give rise to the PJT effect.

To pick examples from perovskite halides, two compounds illustrate this nicely:

- 1) The strong PJT effect (lone pair formation) is the dominant instability in RbGeBr₃, with octahedral tilting secondary. It transforms on cooling from the cubic $Pm\bar{3}m$ aristotype to the polar rhombohedral $R\bar{3}m$ structure shared by CsGeBr₃ to a polar $Pna2_1$ structure which combines tilting with acentric distortions of the Ge²⁺ environment [Z. Anorg. Allg. Chem. 559, 7-16 (1988)].
- 2) The situation is flipped in CsSnBr₃, in which tilting is the dominant instability and the strong PJT effect (lone pair formation) is secondary. It transforms on cooling through the typical cubic and tilted tetragonal and orthorhombic phases with centric Sn²⁺ coordination to two low temperature phases which combine acentric distortions of the Sn²⁺ environment with the extant tilting [arXiv.org: 2401.07978].

The PJT effect and Goldschmidt tolerance factor reflect qualitatively different instabilities. While relatively few compounds which combine both sorts of distortions are known, one instability does not prevent the other. Part of the motivation for this study was precisely to isolate one of these two features at a time: By substituting Pb²⁺ for nearly identically-sized Sr²⁺, we elicit matching phase evolution and similar transition temperatures and lattice parameters, while radically altering the chemistry and bonding so as to essentially eliminate the PJT effect in CsSrBr₃.

Comment 5:

I am not qualified to comment the technical aspects of the theory concerns raised by Reviewer 3. In the context of the experimental findings, I think the theory makes sense and the alignment between the two results is as good as one can hope to find from DFT.

Response: We appreciate the reviewer’s positive assessment of our complementary use of theory and experiment.

On a general note, we are grateful to the reviewer for their critical feedback on our manuscript, which we believe has strengthened our study and its presentation.

Additional notes:

1. The “Author contributions” section has been updated to reflect the new analysis in the Supplementary Information.
2. the VDOS Equation (S10) has been corrected for clarity of notation.

Changes:

It can be shown that the VDOS equals the phonon DOS when the derivatives of the atomic coordinates (velocities, $\mathbf{v}(t)$) are given in terms of the (mass-weighted) normal mode vibrations, *i.e.*, $m_j \mathbf{v}_j(t) = m_j \dot{\mathbf{r}}_j(t) = \sum_s -i\omega_s Q^j(s) e^{-i\omega_s t}$. We can then write the VDOS as the power spectrum of the mass-weighted VACF:

$$\text{VDOS} = \int_0^\infty \frac{\sum_{j=0}^{N_{\text{ions}}} \langle m_j \mathbf{v}_j(t) \cdot m_j \mathbf{v}_j(0) \rangle}{\sum_{j=0}^{N_{\text{ions}}} \langle |m_j \mathbf{v}_j(0)|^2 \rangle} e^{-i\omega t} dt, \quad (3)$$

where $\langle \cdot \rangle$ means the average over all time steps (*i.e.*, shifting reference $t = 0$).

REVIEWER COMMENTS

Reviewer #2 (Remarks to the Author):

The authors have addressed the concerns I brought up previously with the manuscript, and I appreciate their attention to detail with this revision. I have two final suggestions for the manuscript that will help contextualize the findings.

1. Can the authors provide a hypothesis or explanation for why the anharmonicity occurs, even though the PJT effect isn't significant here?
2. There should be a brief discussion in main text reflecting that the DFT calculations do not include spin-orbit coupling, and what this means for the accuracy of the calculations—do the authors expect the DFT results to change in a way that affects the conclusions related to the PJT effect or anharmonicity? I didn't realize that SOC was not included in the calculations when I previously read the manuscript. Many articles have been published (and several are referenced in this manuscript) highlighting the importance of spin-orbit coupling in halide perovskites, so how impactful is this omission?

Various references highlighting the importance of SOC:

- (1) Zhai, Y.; Baniya, S.; Zhang, C.; Li, J.; Haney, P.; Sheng, C.-X.; Ehrenfreund, E.; Vardeny, Z. V. Giant Rashba Splitting in 2D Organic-Inorganic Halide Perovskites Measured by Transient Spectroscopies. *Science Advances* 2017, 3 (7), e1700704. <https://doi.org/10.1126/sciadv.1700704>.
- (2) Even, J.; Pedesseau, L.; Jancu, J. M.; Katan, C. Importance of Spin-Orbit Coupling in Hybrid Organic/Inorganic Perovskites for Photovoltaic Applications. *Journal of Physical Chemistry Letters* 2013, 4 (17), 2999–3005. <https://doi.org/10.1021/jz401532q>.
- (3) Even, J.; Pedesseau, L.; Dupertuis, M.-A.; Jancu, J.-M.; Katan, C. Electronic Model for Self-Assembled Hybrid Organic/Perovskite Semiconductors: Reverse Band Edge Electronic States Ordering and Spin-Orbit Coupling. *Physical Review B* 2012, 86 (20), 205301. <https://doi.org/10.1103/PhysRevB.86.205301>.
- (4) Kepenekian, M.; Even, J. Rashba and Dresselhaus Couplings in Halide Perovskites: Accomplishments and Opportunities for Spintronics and Spin–Orbitronics. *The Journal of Physical Chemistry Letters* 2017, 8 (14), 3362–3370. <https://doi.org/10.1021/acs.jpcllett.7b01015>.
- (5) Zheng, F.; Tan, L. Z.; Liu, S.; Rappe, A. M. Rashba Spin-Orbit Coupling Enhanced Carrier Lifetime in CH₃NH₃PbI₃. *Nano Letters* 2015, 15 (12), 7794–7800. <https://doi.org/10.1021/acs.nanolett.5b01854>.
- (6) Amat, A.; Mosconi, E.; Ronca, E.; Quarti, C.; Umari, P.; Nazeeruddin, M. K.; Grätzel, M.; De Angelis, F. Cation-Induced Band-Gap Tuning in Organohalide Perovskites: Interplay of Spin-Orbit Coupling and Octahedra Tilting. *Nano Letters* 2014, 14 (6), 3608–3616. <https://doi.org/10.1021/nl5012992>.

The manuscript should be published once these two minor issues are addressed.

Reviewer 2:

Comment 1: The authors have addressed the concerns I brought up previously with the manuscript, and I appreciate their attention to detail with this revision. I have two final suggestions for the manuscript that will help contextualize the findings.

Response: We are happy to see that the reviewer is satisfied with our response and appreciate the commitment to resolving our points of disagreement.

Comment 2: Can the authors provide a hypothesis or explanation for why the anharmonicity occurs, even though the PJT effect isn't significant here?

Response: Anharmonicity of the (gerade) tilting motions should be distinguished from anharmonicity of (ungerade) intra-octahedral distortions associated with the PJT effect & lone pair formation. Our finding is that dynamic symmetry breaking due to anharmonic tilting motions causes the broad, nominally forbidden Raman peak, not dynamic symmetry breaking due to vibronic anharmonicity associated with the PJT effect. The tilting is the dominant feature in the majority of compounds in this family that have been studied thus far. For CsPbBr₃ and CsSrBr₃, we showed that anharmonic tilting causes the observed Raman and MD features: we essentially fixed ionic sizes but completely changed the chemical bonding. From this and our additional analysis, it is apparent that anharmonicity from the PJT effect is not the driver for these two compounds. Previously, it appeared the PJT could be the explanation for these observations or at least a contributing factor. We agree that adding a clarifying statement to the discussion can be useful.

Changes:

We augmented the discussion on p. 6 with the following statement:

*Notably, the anharmonicity of the tilting motions is different from that of intra-octahedral distortions associated with the PJT effect.*²⁹

Comment 3:

There should be a brief discussion in main text reflecting that the DFT calculations do not include spin-orbit coupling, and what this means for the accuracy of the calculations—do the authors expect the DFT results to change in a way that affects the conclusions related to the PJT effect or anharmonicity? I didn't realize that SOC was not included in the calculations when I previously read the manuscript. Many articles have been published (and several are referenced in this manuscript) highlighting the importance of spin-orbit coupling in halide perovskites, so how impactful is this omission?

(1) Zhai, Y.; Baniya, S.; Zhang, C.; Li, J.; Haney, P.; Sheng, C.-X.; Ehrenfreund, E.; Vardeny, Z. V. Giant Rashba Splitting in 2D Organic-Inorganic Halide Perovskites Measured by Transient Spectroscopies. *Science Advances* 2017, 3 (7), e1700704. <https://doi.org/10.1126/sciadv.1700704>.

(2) Even, J.; Pedesseau, L.; Jancu, J. M.; Katan, C. Importance of Spin-Orbit Coupling in Hybrid Organic/Inorganic Perovskites for Photovoltaic Applications. *Journal of Physical Chemistry Letters* 2013, 4 (17), 2999–3005. <https://doi.org/10.1021/jz401532q>.

(3) Even, J.; Pedesseau, L.; Dupertuis, M.-A.; Jancu, J.-M.; Katan, C. Electronic Model for Self-Assembled Hybrid Organic/Perovskite Semiconductors: Reverse Band Edge Electronic States Ordering and Spin-Orbit Coupling. *Physical Review B* 2012, 86 (20), 205301. <https://doi.org/10.1103/PhysRevB.86.205301>.

(4) Kepenekian, M.; Even, J. Rashba and Dresselhaus Couplings in Halide Perovskites: Accomplishments and Opportunities for Spintronics and Spin-Orbitronics. *The Journal of Physical*

Chemistry Letters 2017, 8 (14), 3362–3370. <https://doi.org/10.1021/acs.jpcelett.7b01015>.

(5) Zheng, F.; Tan, L. Z.; Liu, S.; Rappe, A. M. Rashba Spin-Orbit Coupling Enhanced Carrier Lifetime in CH₃NH₃PbI₃. Nano Letters 2015, 15 (12), 7794–7800. <https://doi.org/10.1021/acs.nanolett.5b01854>.

(6) Amat, A.; Mosconi, E.; Ronca, E.; Quarti, C.; Umari, P.; Nazeeruddin, M. K.; Grätzel, M.; De Angelis, F. Cation-Induced Band-Gap Tuning in Organohalide Perovskites: Interplay of Spin-Orbit Coupling and Octahedra Tilting. Nano Letters 2014, 14 (6), 3608–3616. <https://doi.org/10.1021/nl5012992>.

Response: The reviewer is correct that it is long-known that spin-orbit coupling (SOC) is an important effect in halide perovskites. It changes the electronic structure in these materials in several ways, i.e., it causes a closing of the band gap or modulation of carrier masses. All of the studies cited by the reviewer focus on electronic-structure properties and confirm the statement that SOC is important for it. Specifically, the article by Amat et al., which is cited by the reviewer as [6] in their comment, found that octahedral tilting causes a change of the band gap that is different depending on whether SOC is accounted for or not. As shown in the paper by Amat et al., this is purely an electronic-structure effect that is rooted in different orbital hybridization that depend on SOC. More relevant for our study, SOC was shown to change quantities related to the total energy and its derivatives in only a minor way. Beck et al. [APL Mater. 7, 021108 (2019)], cited as Ref. 76 in our manuscript, showed DFT calculations yield similar lattice constants and bulk moduli with and without inclusion of SOC. Hence, we expect that including SOC in the calculations would not cause a significant impact on our findings. As recommended by the reviewer, we revised the main text accordingly.

Changes: On p. 7 of the revised article, we now write:

This setup has been shown to accurately describe the structure of HaPs, also in regard to the omission of SOC which impacts electronic-structure properties but does not result in significant changes of quantities related to the total energy.^{75,76}

Comment 4:

The manuscript should be published once these two minor issues are addressed.

Response:

We thank the reviewer again for taking such interest in our work and their helpful comments.

Additional minor changes:

1. We added the following article as ref. 30 to p. 1 of our revised article:
Worhatch, R. J., Kim, H., Swainson, I. P., Yonkeu, A. L. & Billinge, S. J. L. Study of local structure in selected organic–inorganic perovskites in the $Pm\bar{3}m$. Chem. Mater. 20, 1272–1277 (2008)
2. We corrected a typo in the description of the X-ray diffraction experiments on p. 9 of the revised manuscript.

REVIEWERS' COMMENTS

Reviewer #2 (Remarks to the Author):

The manuscript should be published as-is.